# PI(3,5)P$_2$ biosynthesis regulates oligodendrocyte differentiation by intrinsic and extrinsic mechanisms

Yevgeniya A Mironova[1,2], Guy M Lenk[3], Jing-Ping Lin[1], Seung Joon Lee[4], Jeffery L Twiss[4], Ilaria Vaccari[5], Alessandra Bolino[5], Leif A Havton[6], Sang H Min[7], Charles S Abrams[7], Peter Shrager[8], Miriam H Meisler[3,9], Roman J Giger[1,9]*

[1]Department of Cell and Developmental Biology, University of Michigan School of Medicine, Ann Arbor, United States; [2]Cellular and Molecular Biology Graduate Program, University of Michigan School of Medicine, Ann Arbor, United States; [3]Department of Human Genetics, University of Michigan School of Medicine, Ann Arbor, United States; [4]Department of Biological Sciences, University of South Carolina, Columbia, United States; [5]Human Inherited Neuropathies Unit, INSPE-Institute for Experimental Neurology, San Raffaele Scientific Institute, Milan, Italy; [6]Department of Neurology, David Geffen School of Medicine at UCLA, Los Angeles, United States; [7]Department of Medicine, University of Pennsylvania School of Medicine, Philadelphia, United States; [8]Department of Neurobiology and Anatomy, University of Rochester Medical Center, Rochester, United States; [9]Department of Neurology, University of Michigan School of Medicine, Ann Arbor, United States

*For correspondence: rgiger@med.umich.edu

Competing interests: The authors declare that no competing interests exist.

**Abstract** Proper development of the CNS axon-glia unit requires bi-directional communication between axons and oligodendrocytes (OLs). We show that the signaling lipid phosphatidylinositol-3,5-bisphosphate [PI(3,5)P$_2$] is required in neurons and in OLs for normal CNS myelination. In mice, mutations of *Fig4, Pikfyve* or *Vac14,* encoding key components of the PI(3,5)P$_2$ biosynthetic complex, each lead to impaired OL maturation, severe CNS hypomyelination and delayed propagation of compound action potentials. Primary OLs deficient in *Fig4* accumulate large LAMP1$^+$ and Rab7$^+$ vesicular structures and exhibit reduced membrane sheet expansion. PI(3,5)P$_2$ deficiency leads to accumulation of myelin-associated glycoprotein (MAG) in LAMP1$^+$perinuclear vesicles that fail to migrate to the nascent myelin sheet. Live-cell imaging of OLs after genetic or pharmacological inhibition of PI(3,5)P$_2$ synthesis revealed impaired trafficking of plasma membrane-derived MAG through the endolysosomal system in primary cells and brain tissue. Collectively, our studies identify PI(3,5)P$_2$ as a key regulator of myelin membrane trafficking and myelinogenesis.

## Introduction

In the vertebrate CNS, the majority of long axons are myelinated. Myelin greatly increases the conduction velocity of action potentials and provides metabolic support for axons. Bidirectional axo-glial signaling is critical for nervous system myelination and fiber stability (*Nave and Trapp, 2008*; *Simons and Lyons, 2013*). Myelin development is regulated by oligodendrocyte (OL) intrinsic mechanisms (*Zuchero and Barres, 2013*), astrocyte secreted factors (*Ishibashi et al., 2006*), neuronal electrical activity (*Barres and Raff, 1993*; *Ishibashi et al., 2006*) and axon derived chemical signals (*Coman et al., 2005*; *Ohno et al., 2009*; *Winters et al., 2011*, *Yao et al., 2014*).

**eLife digest** Neurons communicate with each other through long cable-like extensions called axons. An insulating sheath called myelin (or white matter) surrounds each axon, and allows electrical impulses to travel more quickly. Cells in the brain called oligodendrocytes produce myelin. If the myelin sheath is not properly formed during development, or is damaged by injury or disease, the consequences can include paralysis, impaired thought, and loss of vision.

Oligodendrocytes have complex shapes, and each can generate myelin for as many as 50 axons. Oligodendrocytes produce the building blocks of myelin inside their cell bodies, by following instructions encoded by genes within the nucleus. However, the signals that regulate the trafficking of these components to the myelin sheath are poorly understood.

Mironova et al. set out to determine whether signaling molecules called phosphoinositides help oligodendrocytes to mature and move myelin building blocks from the cell bodies to remote contact points with axons. Genetic techniques were used to manipulate an enzyme complex in mice that controls the production and turnover of a phosphoinositide called $PI(3,5)P_2$. Mironova et al. found that reducing the levels of $PI(3,5)P_2$ in oligodendrocytes caused the trafficking of certain myelin building blocks to stall. Key myelin components instead accumulated inside bubble-like structures near the oligodendrocyte's cell body. This showed that $PI(3,5)P_2$ in oligodendrocytes is essential for generating myelin. Further experiments then revealed that reducing $PI(3,5)P_2$ in the neurons themselves indirectly prevented the oligodendrocytes from maturing. This suggests that $PI(3,5)P_2$ also takes part in communication between axons and oligodendrocytes during development of the myelin sheath.

A key next step will be to identify the regulatory mechanisms that control the production of $PI(3,5)P_2$ in oligodendrocytes and neurons. Future studies could also explore what $PI(3,5)P_2$ acts upon inside the axons, and which signaling molecules support the maturation of oligodendrocytes. Finally, it remains unclear whether $PI(3,5)P_2$ signaling is also required for stabilizing mature myelin, and for repairing myelin after injury in the adult brain. Further work could therefore address these questions as well.

Disorders associated with defective CNS white matter range from multiple sclerosis and inherited leukodystrophies to psychiatric disorders (*Fields, 2008*; *Makinodan et al., 2012*; *Perlman and Mar, 2012*).

FIG4 is an evolutionarily conserved lipid phosphatase that removes the 5' phosphate group from phosphatidylinositol(3,5)bisphosphate [$PI(3,5)P_2$] to produce PI(3)P. Together with its antagonistic kinase PIKFYVE and the scaffold protein VAC14, FIG4 forms an enzyme complex that regulates the interconversion of PI(3)P and $PI(3,5)P_2$ on membranes of the late endosomal/ lysosomal (LE/Lys) compartment (*Jin et al., 2008*; *McCartney et al., 2014*). In addition to its 5'-phosphatase activity, Fig4 is required to stabilize the enzyme complex. $PI(3,5)P_2$ directly regulates the lysosomal cation channels TRPML1, TPC1 and TPC2 (*Dong et al., 2010*; *Wang et al., 2012*; *2014*). Reduced activity of these lysosomal channels and the resulting osmotic enlargement of the LE/Lys may underlie vacuolization in *Fig4* null cells (*Lenk and Meisler, 2014*). Consistent with this model, overexpression of TRPML1 in *Vac14* and *Fig4* mutant cells appears to rescue vacuolization (*Dong et al., 2010*; *Zou et al., 2015*). In *Drosophila*, loss of TRPML1 generates a muscle vacuolization phenotype reminiscent of FIG4 deficiency (*Bharadwaj et al., 2016*).

FIG4 deficiency is particularly harmful for neural cells with elaborate morphologies, including projection neurons and myelinating glia. Mutations of human *FIG4* result in neurological disorders including Charcot-Marie-Tooth type 4J, a severe form of peripheral neuropathy (*Chow et al., 2007*; *Nicholson et al., 2011*), polymicrogyria with epilepsy (*Baulac et al., 2014*), and Yunis-Varon syndrome (*Campeau et al., 2013*). Mice null for *Fig4* exhibit severe tremor, brain region-specific spongiform degeneration, hypomyelination, and juvenile lethality (*Chow et al., 2007*; *Ferguson et al., 2009*; *Winters et al., 2011*). We previously demonstrated that a *Fig4* transgene driven by the neuron-specific enolase (*NSE*) promoter rescued juvenile lethality and neurodegeneration in global *Fig4* null mice, and that these phenotypes were not rescued by an astrocyte-specific *Fig4* transgene

(*Ferguson et al., 2012*). The neuron-specific transgene also rescued conduction in peripheral nerves (*Ferguson et al., 2012*) and structural defects in CNS myelination (*Winters et al., 2011*). Conversely, inactivation of *Fig4* specifically in neurons resulted in region-specific neurodegeneration (*Ferguson et al., 2012*).

The cellular and molecular mechanisms relating loss of *Fig4* to hypomyelination are poorly understood. To further characterize the requirement of PI(3,5)P$_2$ for CNS myelination, we manipulated individual components of the PI(3,5)P$_2$ biosynthetic complex. *Pikfyve* and *Vac14* global null mice die prematurely, before the onset of CNS myelination (*Zhang et al., 2007*; *Ikonomov et al., 2011*). To circumvent this limitation, we employed a combination of conditional null alleles and hypomorphic alleles in the mouse. Our study shows that multiple strategies to perturb the FIG4/PIKFYVE/VAC14 enzyme complex, and by extension the lipid product PI(3,5)P$_2$, result in the common endpoints of arrested OL differentiation, impaired myelin protein trafficking through the LE/Lys compartment, and severe CNS hypomyelination. We demonstrate that these defects in myelin biogenesis are functionally relevant and result in faulty conduction of electrical impulses.

## Results

### Conditional ablation of *Fig4* in neurons or the OL lineage results in CNS hypomyelination

In the early postnatal brain, *Fig4* is broadly expressed and enriched in oligodendrocyte progenitor cells (OPCs) and newly formed OLs (NFOs) (*Zhang et al., 2014*). Mice in which exon 4 of the *Fig4* gene is flanked by *loxP* sites (*Ferguson et al., 2012*) were used to generate *Fig4$^{-/flox}$,SynCre* and *Fig4$^{-/flox}$, Olig2Cre* mice deficient for *Fig4* in neurons or OLs, respectively. Myelin development in these conditional mutants, as well as the *Fig4* global mutant (*Fig4$^{-/-}$*) and control mice (*Fig4$^{+/+}$* and *Fig4$^{flox/+}$*), was analyzed by Fluoromyelin Green labeling (*Figure 1*). In control brains, the corpus callosum and internal capsule were prominently labeled (*Figure 1A and A'*). Staining of these structures was weaker in *Fig4$^{-/flox}$,SynCre* brains and further reduced in *Fig4$^{-/flox}$,Olig2Cre* and *Fig4$^{-/-}$* brains (*Figure 1B-D'*). For a quantitative comparison of the myelination defects, whole brain membranes were prepared from P21 pups and analyzed by immunoblotting with antibodies specific for the myelin markers myelin-associated glycoprotein (MAG), 2′,3′-cyclic-nucleotide 3′-phosphodiesterase (CNPase), proteolipid protein (PLP), and myelin basic protein (MBP) (*Figure 1E*). Compared to *Fig4$^{+/+}$* membranes, a significant reduction in myelin proteins was evident in *Fig4$^{-/-}$* mice, *Fig4$^{-/flox}$,SynCre* mice and *Fig4$^{-/flox}$,Olig2Cre* mice (*Figure 1F -I*). The finding that the neuronal marker classIII *β*-tubulin is not significantly decreased in any of these mice indicates that the decrease in CNS myelin is not secondary to neuronal loss. While the *Olig2* promoter is highly active in the OL lineage, activity has also been reported in astrocytes and a subset of neurons (*Dessaud et al., 2007*; *Zhang et al., 2014*). To independently assess the role of *Fig4* in the OL lineage, we generated *Fig4$^{-/flox}$,PdgfraCreER* mice that permit tamoxifen inducible gene ablation. At postnatal-days (P)5 and 6, before the onset of CNS myelination, *Fig4$^{-/flox}$,PdgfraCreER* pups were injected with 4-hydroxytamoxifen and brains were analyzed at P20-P21. Inducible ablation of *Fig4* in the OL-linage resulted in reduced expression of the myelin proteins CNPase, MAG, and MBP, as assessed by Western blot analysis (*Figure 1—figure supplement 1A–B'*) as well as myelin loss in forebrain structures and cerebellar white matter (*Figure 1—figure supplement 1C–D'*). Fewer *Plp1$^+$* OLs were present in optic nerve sections of *Fig4$^{-/flox}$,PdgfraCreER* mice (*Figure 1—figure supplement 1E and E'*). Together, these studies indicate that proper CNS myelination is dependent upon OL cell-autonomous (intrinsic) functions of *Fig4,* in addition to non-OL-autonomous (extrinsic) functions of *Fig4* provided by neurons.

As previously described, *Fig4$^{-/flox}$,SynCre* mice exhibit impaired movement and region-specific vacuolization and neurodegeneration (*Figure 1—figure supplement 2A'',B'',C'',D''*) (*Ferguson et al., 2012*). In contrast, *Fig4$^{-/flox}$,Olig2Cre* mice exhibit very mild vacuolization in brain (*Figure 1—figure supplement 2A''',B''',C''',D'''*). Consistent with the known expression of the *Olig2* promoter in motor neurons (*Mizuguchi et al., 2001*) ventral spinal cord of *Fig4$^{-/flox}$,Olig2Cre* mice shows extensive vacuolization (*Figure 1—figure supplement 2D'''*), similar to *Fig4$^{-/flox}$, Mnx1Cre* (otherwise referred to as *Fig4$^{-/flox}$,Hb9Cre*) mice (*Figure 1—figure supplement 2E*) (*Vaccari et al., 2015*). Analysis of *Fig4$^{-/flox}$,Hb9Cre* spinal cord identified enlarged vacuoles within motoneuron axons, greatly extending their diameter and pushing the axoplasm into a thin peripheral

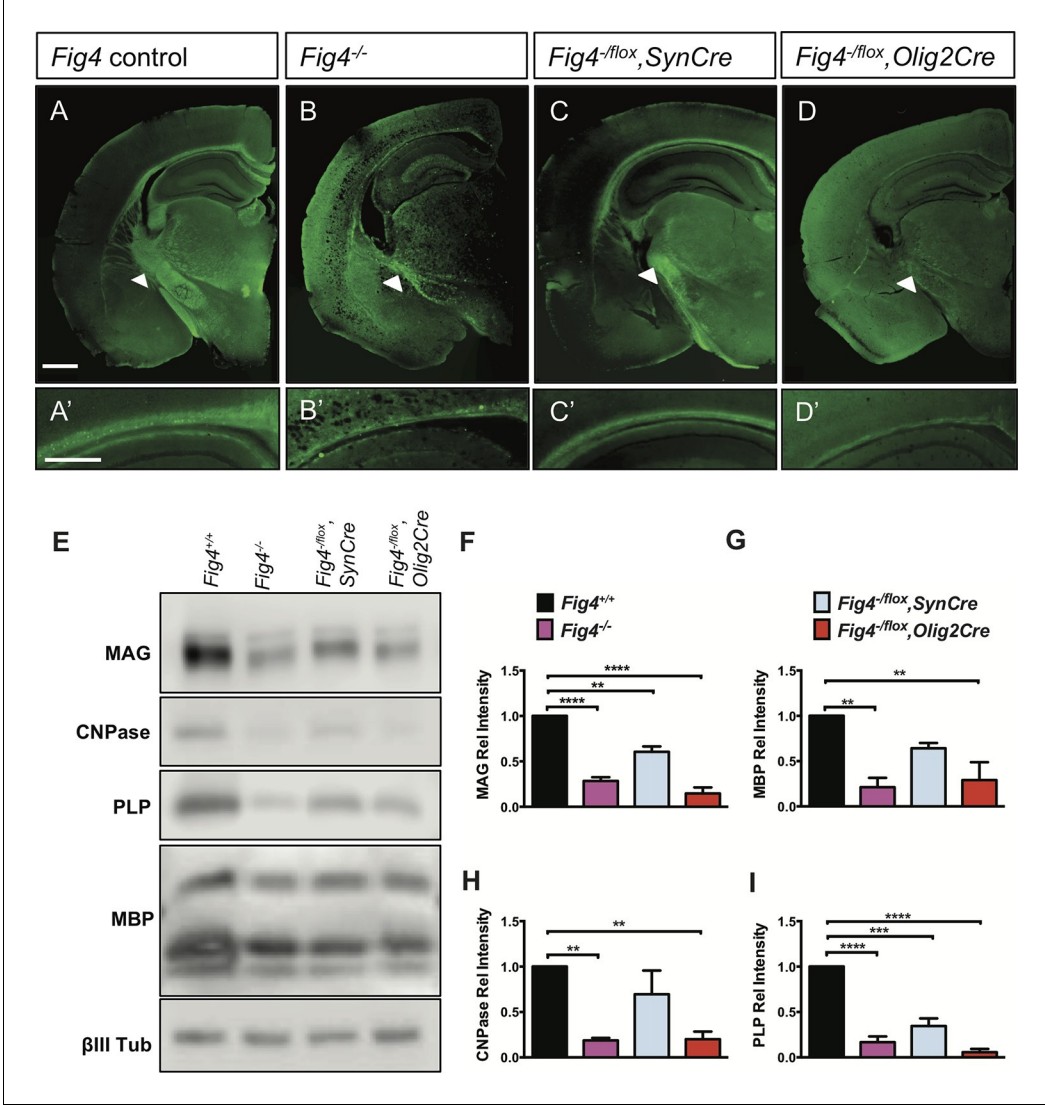

**Figure 1.** Conditional ablation of *Fig4* in neurons or OLs leads to CNS hypomyelination. (**A-D**) Coronal sections of juvenile (P21-30) mouse forebrain stained with FluoroMyelin Green. (**A**) *Fig4* control mice (harboring at least one *Fig4* WT allele), (**B**) *Fig4* germline null mice (*Fig4*$^{-/-}$), (**C**) *Fig4*$^{-/flox}$,*SynCre* mice and (**D**) *Fig4*$^{-/flox}$,*Olig2Cre* mice. Thinning of the corpus callosum and internal capsule (white arrowheads) is observed in *Fig4*$^{-/-}$, *Fig4*$^{-/flox}$,*SynCre*, and *Fig4*$^{-flox}$,*Olig2Cre* mice. (**A'-D'**) Higher magnification images of the corpus callosum. Scale bar (**A-D**), 1 mm and (**A'-D'**), 400 µm. (**E**) Representative Western blots of P21 brain membranes prepared from *Fig4*$^{+/+}$ (WT), *Fig4*$^{-/-}$, *Fig4*$^{-/flox}$,*SynCre* and *Fig4*$^{-/flox}$,*Olig2Cre* mice probed with antibodies specific for the myelin proteins MAG, CNPase, PLP, and MBP. To control for protein loading, membranes were probed for the neuronal marker class III β-tubulin (βIII Tub). (**F-I**) Quantification of Western blot signals for MAG, MBP, CNPase, and PLP in *Fig4*$^{+/+}$ (black bars), *Fig4*$^{-/-}$ (purple bars), *Fig4*$^{-/flox}$,*SynCre* (light blue bars), and *Fig4*$^{-flox}$,*Olig2Cre* (red bars) brain membranes. Quantification of myelin protein signals is normalized to βIII Tub. Relative protein intensities compared to WT brain are shown as mean value ± SEM. For each of the four genotypes, three independent membrane preparations were carried out. One-way ANOVA with multiple comparisons, Dunnett posthoc test; **p<0.01, ***p<0.001 and ****p<0.0001. An independent strategy for OL-specific *Fig4* deletion results in a similar phenotype as shown in *Figure 1—figure supplement 1*. Histochemical staining of brain, spinal cord and dorsal root ganglion tissue sections of *Fig4* conditional knock-out mice, as well as Kaplan-Meier plots for *Fig4*$^{-/flox}$,*SynCre* and *Fig4*$^{-flox}$, *Olig2Cre* mice are shown in *Figure 1—figure supplement 2*.

The following figure supplements are available for figure 1:

**Figure supplement 1.** CNS hypomyelination in *Fig4*$^{-/flox}$,*PdgfαCreER* mice.

*Figure 1 continued on next page*

*Figure 1 continued*

**Figure supplement 2.** Loss of *Fig4* in the OL-lineage or neurons differentially affects spongiform degeneration and lifespan.

rim near the plasma membrane (*Figure 1—figure supplement 2F*). In contrast to the movement disability and reduced survival of *Fig4*$^{-/flox}$,*SynCre* mice,(*Ferguson et al., 2012*) the movement of *Fig4*$^{-/flox}$,*Olig2Cre* mice is normal and no premature death was observed, with the oldest now surviving beyond 14 months of age (*Figure 1—figure supplement 2G*). There are no obvious defects in mobility of littermate controls and *Fig4*$^{-/flox}$,*Olig2Cre* conditional mutant mice at P23, as demonstrated in the *Videos 1* and *2*.

## *Fig4* deficiency in neurons or OLs leads to developmental dysmyelination of the optic nerve

Analysis of P21 retina revealed the presence of numerous vacuoles in the inner retina of *Fig4*$^{-/flox}$,*SynCre* mice but no defects in overall morphology or stratification (*Figure 2A'*). No vacuoles were detected in the *Fig4*$^{-/flox}$,*Olig2Cre* retina (*Figure 2A''*). For ultrastructural analysis, optic nerves of *Fig4* conditional knock-out mice were processed for transmission electron microscopy (TEM). In P21 *Fig4* control mice (retaining at least one intact allele of *Fig4*), the fraction of myelinated axons in the optic nerve is 79± 2%. In optic nerves of *Fig4*$^{-/flox}$,*SynCre* mice, only 9± 3% of axons are myelinated and in *Fig4*$^{-/flox}$,*Olig2Cre* mice only 12± 1% of axons are myelinated (*Figure 2B–B'' and D*). To assess myelin health, we determined the g-ratio (the ratio of the inner axonal diameter to the total fiber diameter) of myelinated axons in the optic nerve of *Fig4* control and conditional mutants. Compared to control mice, a small but significant increase in g-ratio was observed in *Fig4*$^{-/flox}$,*SynCre* and *Fig4*$^{-/flox}$,*Olig2Cre* mice, an indication of myelin thinning (*Figure 2E*). To determine whether the optic nerve hypomyelination at P21 reflects a transient delay in myelin development, rather than a lasting defect, we repeated the analysis with adult mice. Similar to P21 optic nerves, ultrastructural analysis of both types of adult optic nerves revealed profound hypomyelination (*Figure 2C–C''*). At P60-75, 92± 2% of axons are myelinated in *Fig4* control nerves. This is reduced to 16± 4% in *Fig4*$^{-/flox}$,*SynCre* mice and 12± 2% in *Fig4*$^{-/flox}$,*Olig2Cre* mice (*Figure 2F*). It is noteworthy that conditional ablation of *Fig4* either in neurons or OLs leads to preferential absence of myelin sheaths on small and intermediate caliber axons, while many large caliber axons undergo myelination (*Figure 2B',B'',C' and C''*).

Few axons in the optic nerve of adult *Fig4*$^{-/flox}$,*SynCre* mice showed signs of degeneration (*Figure 2C'*). No evidence for axonal degeneration was observed in *Fig4*$^{-/flox}$,*Olig2Cre* optic nerves. CNS hypomyelination in *Fig4*$^{-/flox}$,*Olig2Cre* mice was still present at P150, the oldest time point examined by TEM (data not shown). Thus, the optic nerve hypomyelination observed at P21 is not transient in nature but persists into adulthood. We conclude that selective ablation of *Fig4* either in neurons or in the OL lineage leads to profound CNS dysmyelination.

## Conditional ablation of *Fig4* in neurons or the OL lineage impairs nerve conduction

To determine whether the morphological defects in CNS myelin of *Fig4* conditional mutants result in functional deficits, we performed electrophysiological recordings. We measured the conduction velocity and amplitude of compound action potentials (CAPs) in optic nerves acutely isolated from P21 mice. Global deletion of *Fig4* (*Fig4*$^{-/-}$) results in a dramatic reduction in a population of fast conducting fibers and a corresponding increase in the proportion of slowly conducting fibers (*Figure 3A,B,E*) (*Winters et al., 2011*). The average velocity of the largest peak in *Fig4* control nerves carrying at least one intact allele of *Fig4* is 1.9 ± 0.1 m/s but in *Fig4*$^{-/-}$ nerves this is reduced to 0.7 ± 0.2 m/s. A similar CAP redistribution was observed in optic nerves prepared from *Fig4*$^{-/flox}$, *SynCre* mice (0.7 ± 0.1 m/s) and *Fig4*$^{-/flox}$,*Olig2Cre* mice (0.6 ± 0.03 m/s) (*Figure 3C,D,E*). Thus, consistent with biochemical and morphological analyses (*Figures 1* and *2*), loss of *Fig4* in neurons or in the OL-lineage results in slowed nerve conduction.

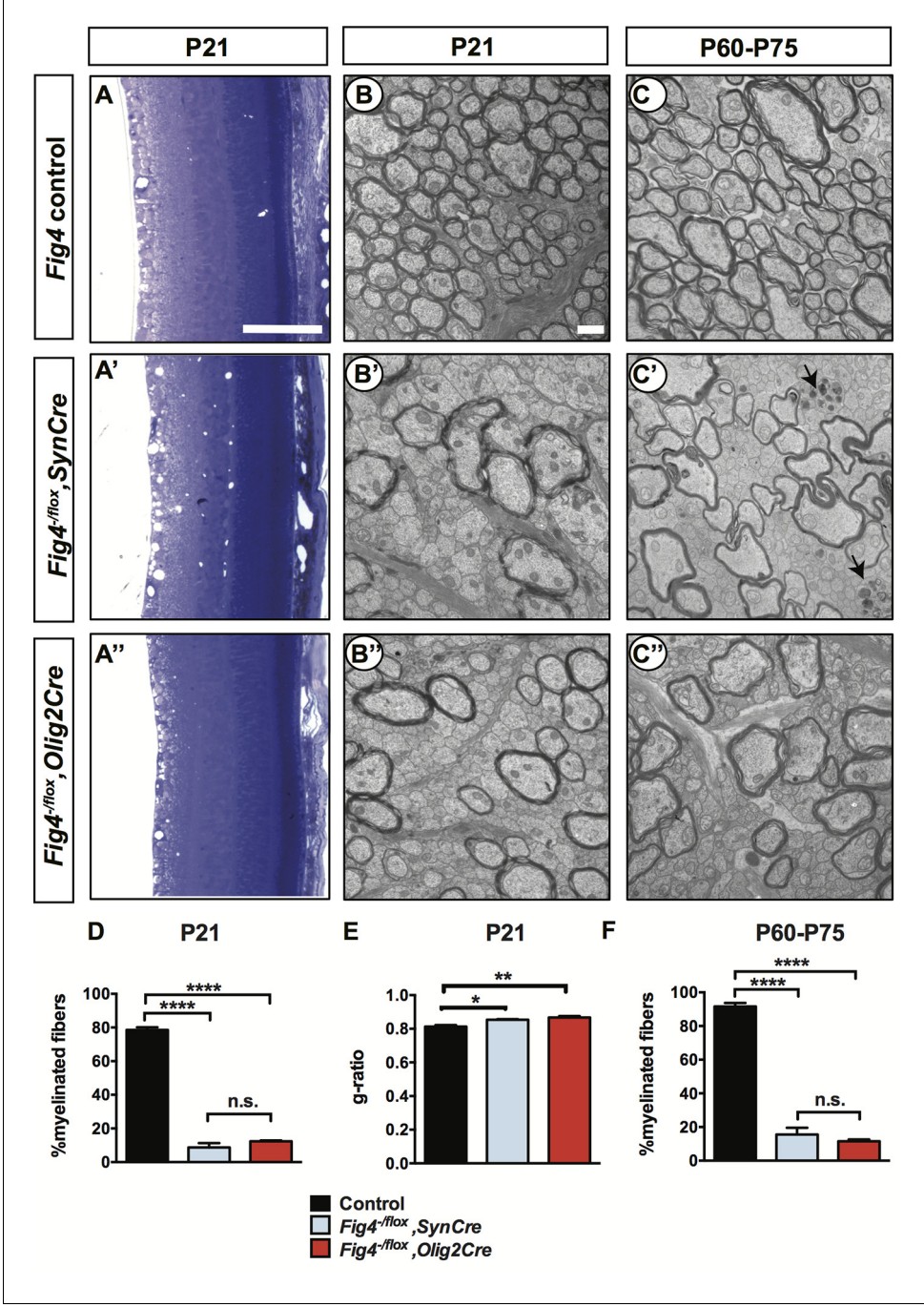

**Figure 2.** Conditional ablation of *Fig4* in neurons or in OLs leads to severe dysmyelination of the optic nerve. (A-A'') Sagittal sections of juvenile (P21) mouse retina embedded in epoxy resin and stained with toluidine blue. (A) *Fig4* control mice, harboring at least one *Fig4* WT allele, (A') *Fig4*<sup>-/flox</sup>,*SynCre* mice and (A'') *Fig4*<sup>-/flox</sup>,*Olig2Cre* mice. Scale bar, 100 µm. (B-B'') Representative TEM images of optic nerve cross sections of P21 (B) *Fig4* control, (B') *Fig4*<sup>-/flox</sup>,*SynCre* and (B'') *Fig4*<sup>-/flox</sup>,*Olig2Cre* mice. (C-C'') Representative TEM images of optic nerve cross sections of adult (P60-75) mice. (C) *Fig4* control, (C') *Fig4*<sup>-/flox</sup>,*SynCre* and (C'') *Fig4*<sup>-/flox</sup>,*Olig2Cre* mice. Black arrows in C' indicate the presence of dystrophic axons. Scale bar (B-C'') = 1 µm. (D) Quantification of percentage of myelinated fibers in the optic nerve at P21 and P60-75. At P21, *Fig4* controls (n = 3 mice, 3 nerves); *Fig4*<sup>-/flox</sup>, *SynCre* (n = 2 mice, 3 nerves) and *Fig4*<sup>-/flox</sup>,*Olig2Cre* (n = 3 mice, 3 nerves). (E) Quantification of myelinated fiber g-ratios in the optic nerve at P21, n = 3 animals, 3 nerves for all groups. (F) Quantification of myelinated fibers in the optic nerve at P60-P75. *Fig4* control (n = 4 mice, 4 nerves), *Fig4*<sup>-/flox</sup>,*SynCre* (n = 4 mice, 4 nerves); *Fig4*<sup>-/flox</sup>,

*Figure 2 continued on next page*

*Figure 2 continued*

*Olig2Cr*e (n = 3 mice, 4 nerves). Results are shown as mean value ± SEM, one-way ANOVA with multiple comparisons, Tukey posthoc test; n.s. p>0.05, *p=0.0211, **p=0.0055, ****p<0.0001.

## Reduced number of mature OLs in *Fig4⁻ᐟ<sup>flox</sup>,Olig2Cre* and *Fig14⁻ᐟ<sup>flox</sup>, SynCre* optic nerves

To assess the cellular basis of the CNS hypomyelination phenotype, we stained optic nerve cross sections from *Fig4* conditional mutants for markers in the OL lineage. Compared to *Fig4* control optic nerves, the diameter of nerves from P21 *Fig4⁻ᐟflox,SynCre* and *Fig4⁻ᐟflox,Olig2Cre* mice were each reduced by 20%. The density of $NG2^+$ progenitor cells in optic nerve tissue sections is comparable among the three genotypes (**Figure 4A–A'' and D**). The density of $Olig2^+$ cells, a marker that labels immature and mature OLs, is reduced, as is labeling of *Plp1*, a mature OL marker (**Figure 4B-B'',C–C'',E and F**). These studies indicate that OPCs are present at normal density and tissue distribution in the *Fig4* conditional null optic nerves, but they fail to generate the normal population of mature myelin-forming OLs.

## Loss of *Fig4* attenuates OL differentiation in vitro

For a more detailed analysis of the OL lineage, we isolated primary OPCs from P6-P14 *Fig4* pups by anti-PDGFRα immunopanning (**Emery and Dugas, 2013**). Yields of OPCs per brain did not differ between control and *Fig4*-deficient mice (data not shown). OPCs were cultured for two days in vitro (DIV2) under proliferating conditions, fixed and analyzed by double-immunofluorescence staining of Ki67 and PDGFRα. The density of $Ki67^+/PDGFRα^+$ cells in $Fig4^{+/+}$ and $Fig4^{-/-}$ cultures is very similar

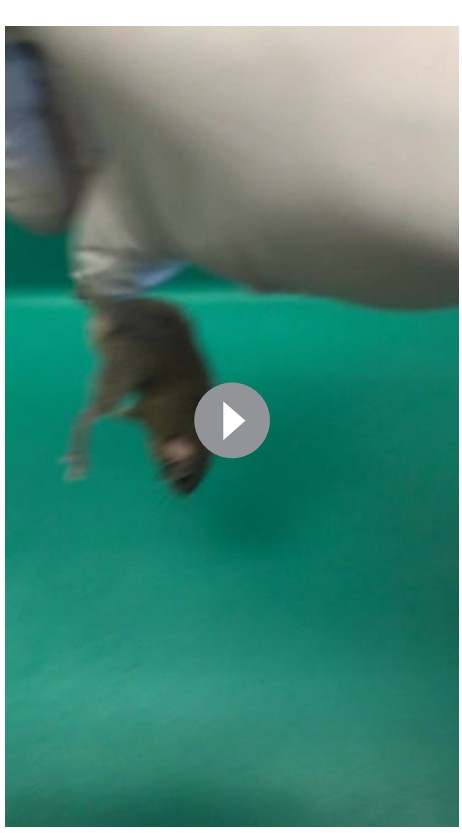

**Video 1.** Normal locomotion of juvenile $Fig4^{+/flox}$, *Olig2Cre* mice. A representative video of a control $Fig4^{+/flox}$,*Olig2Cre* mouse at P23. N = 10

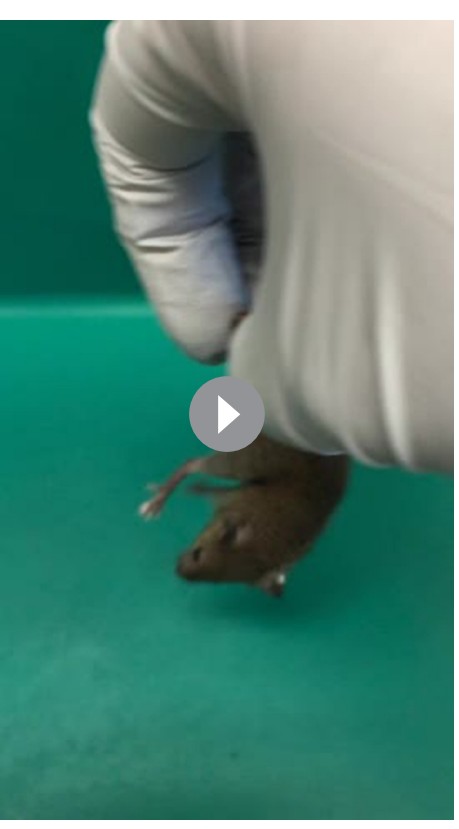

**Video 2.** Normal locomotion of juvenile $Fig4^{-/flox}$, *Olig2Cre* mice. A representative video of a $Fig4^{-/flox}$, *Olig2Cre* conditional mutant mouse at P23 shows no obvious pathology in locomotion. N = 10

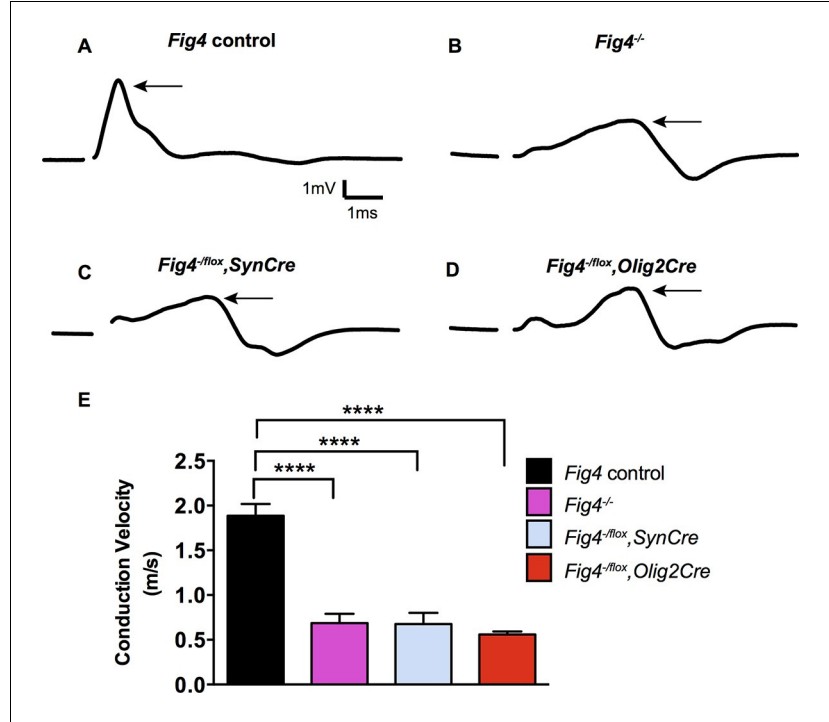

**Figure 3.** Conditional ablation of *Fig4* in neurons or OLs leads to impaired conduction of electrical impulses in the optic nerve. Compound action potential (CAP) recordings from acutely isolated optic nerves of P21 mice. (**A**) Representative CAP traces recorded from *Fig4* control mice, harboring at least one *Fig4* WT allele (n = 14 nerves), (**B**) *Fig4*$^{-/-}$ mice (n = 5 nerves), (**C**) *Fig4*$^{-/flox}$,*SynCre* mice (n = 11 nerves) and (**D**) *Fig4*$^{-/flox}$,*Olig2Cre* mice (n = 9 nerves). For each graph, the arrow indicates the largest amplitude peak, as identified by Gaussian fit. (**E**) Quantification of average conduction velocity of largest amplitude peaks identified in **A-D**. Results are shown as mean value ± SEM, one-way ANOVA with multiple comparisons, Dunnett posthoc, ****$p < 0.0001$.

(*Figure 5—figure supplement 1A–B*). After culture under standard differentiation conditions for 4 days, absence of PDGF and presence of triiodothyronine (T3), OPCs isolated from *Fig4*$^{+/+}$ (control) or *Fig4*$^{-/-}$ pups both acquire a highly arborized morphology and positive staining for OL markers. The density of NG2$^+$ cells and CNPase$^+$ cells, normalized to Hoechst 33,342 dye$^+$ nuclei, is comparable among wildtype and *Fig4*-deficient cultures (*Figure 5A–B', and C*). However, the fraction of cells expressing the more mature OL markers MAG and MBP was significantly reduced in *Fig4*$^{-/-}$ cultures (*Figure 5A–B', and C*). A more detailed categorization of post-mitotic OLs, based on actin and MBP double-labeling, revealed a significantly decreased number of *Fig4*-deficient OLs that matured to a stage with lamellar MBP$^+$ membrane sheets (*Zuchero et al., 2015*) (*Figure 5D–E*). The reduced number of mature OLs in *Fig4*$^{-/-}$ cultures was not caused by increased cell death (*Figure 5—figure supplement 1C–E*). For a quantitative assessment of protein expression in primary OLs from *Fig4*$^{+/+}$ and *Fig4*$^{-/-}$ brains, DIV 3 cultures were lysed and analyzed by capillary Western blotting (*Figure 5—figure supplement 2A–C*). FIG4 is clearly detected in *Fig4*$^{+/+}$ OL lysates but not in *Fig4*$^{-/-}$ OL lysates. In *Fig4*$^{-/-}$ lysates MAG is significantly reduced. Collectively, these data suggest that the initial programs of OL maturation progress normally in the absence of *Fig4* while later stages of OL-differentiation, including lamellar membrane expansion, are *Fig4*-dependent.

## Independent perturbation of three components of the PI(3,5)P$_2$ biosynthetic complex all result in severe CNS hypomyelination

Together with the kinase PIKFYVE and the scaffolding protein VAC14, FIG4 forms a biosynthetic complex necessary for acute interconversion of PI(3) and PI(3,5)P$_2$. The complex is located on the cytosolic surface of vesicles trafficking through the LE/Lys compartment (*McCartney et al., 2014*). As an independent test of the effect of perturbation of the FIG4/PIKFYVE/VAC14 enzyme complex

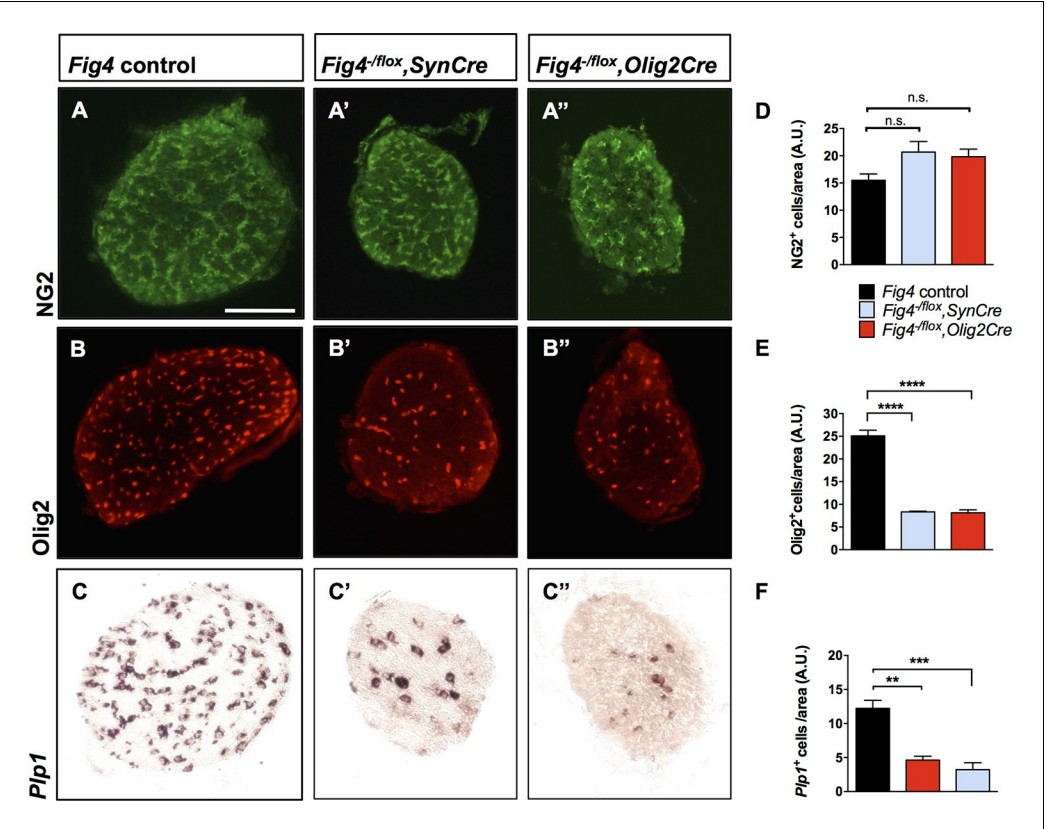

**Figure 4.** Conditional ablation of *Fig4* in neurons or OLs results in a decrease of mature OLs. (**A, B, C**) Optic nerve cross sections from P21 *Fig4* control mice, harboring at least one *Fig4* WT allele, (**A', B', C'**) *Fig4*$^{-/flox}$,*SynCre* mice and (**A'', B'', C''**) *Fig4*$^{-/flox}$,*Olig2Cre* mice were stained with anti-NG2, anti-Olig2 or probed for *Plp1* mRNA expression. Scale bar = 100 μm. (**D-F**) Quantification of labeled cells in optic nerve cross sections normalized to area in arbitrary units (A.U.). (**D**) The density of NG2$^+$ cells in *Fig4* control (n = 4 mice), *Fig4*$^{-/flox}$,*SynCre* (n = 3 mice) and *Fig4*$^{-/flox}$,*Olig2Cre* (n = 3 mice) optic nerves is not significantly (n.s.) different. (**E**) Quantification of the density of Olig2$^+$ cells in *Fig4* control (n = 6 mice), *Fig4*$^{-/flox}$,*SynCre* (n = 3 mice) and *Fig4*$^{-/flox}$,*Olig2Cre* (n = 4 mice) optic nerves. (**F**) Quantification of the density of *Plp1*$^+$ cells in *Fig4* control (n = 8 mice), *Fig4*$^{-/flox}$,*SynCre* (n = 4 mice) and *Fig4*$^{-/flox}$,*Olig2Cre* (n = 4 mice) optic nerves. Results are shown as mean value ± SEM, one-way ANOVA with multiple comparisons, Dunnett's posthoc test. **$p=0.001$, ***$p=0.0002$ and ****$p<0.0001$.

on CNS myelination, we generated *Pikfyve*$^{flox/flox}$,*Olig2cre* mice predicted to be more severely deficient in PI(3,5)P$_2$ than the FIG4 and VAC14 mutants. Consistent with this expectation, the phenotype of the *Pikfyve* mutant mice is much more severe, with a significant tremor (*Videos 3* and *4*) and death at 2 weeks of age (n = 16 pups). FluoroMyelin Green staining of P13 brain tissue revealed profound hypomyelination of the corpus callosum, internal capsule and cerebellar white matter of *Pikfyve*$^{flox/flox}$,*Olig2cre* pups (*Figure 6A–A'*). In situ hybridization of *Plp1* revealed a virtual absence of mature OLs in the *Pikfyve*$^{flox/flox}$,*Olig2cre* brain, including structures in the forebrain and cerebellar white matter (*Figure 6B–D'*). Toluidine blue staining of P13 optic nerve sections revealed many fibers with clearly visible myelin profiles in *Pikfyve* positive control mice and a striking absence of myelin profiles in *Pikfyve*$^{flox/flox}$;*Olig2cre* conditional mutants (*Figure 6—Supplement 1B–D*). Moreover, deficiency of *Pikfyve* in OLs results in a pronounced accumulation of large perinuclear vesicles in the optic nerve (*Figure 6—figure supplement 1B,D*). Defects in differentiation of *Pikfyve*$^{-/-}$ OL cultures are more pronounced than in *Fig4*$^{-/-}$ OL cultures. Deficiency of *Pikfyve* reduces OPC proliferation (*Figure 6E–E' and G*) and results in a 95 ± 1% reduction in cells that progress to the MBP$^+$ stage, compared with wildtype cells (*Figure 6F–F' and H*). In addition to *Fig4* and *Pikfyve* mutants, we also examined myelinogenesis in the well-characterized recessive *Vac14* mouse mutant L156R (*Vac14*$^{L156R}$) (*Jin et al., 2008*). The L156R missense mutation impairs the interaction of VAC14 with

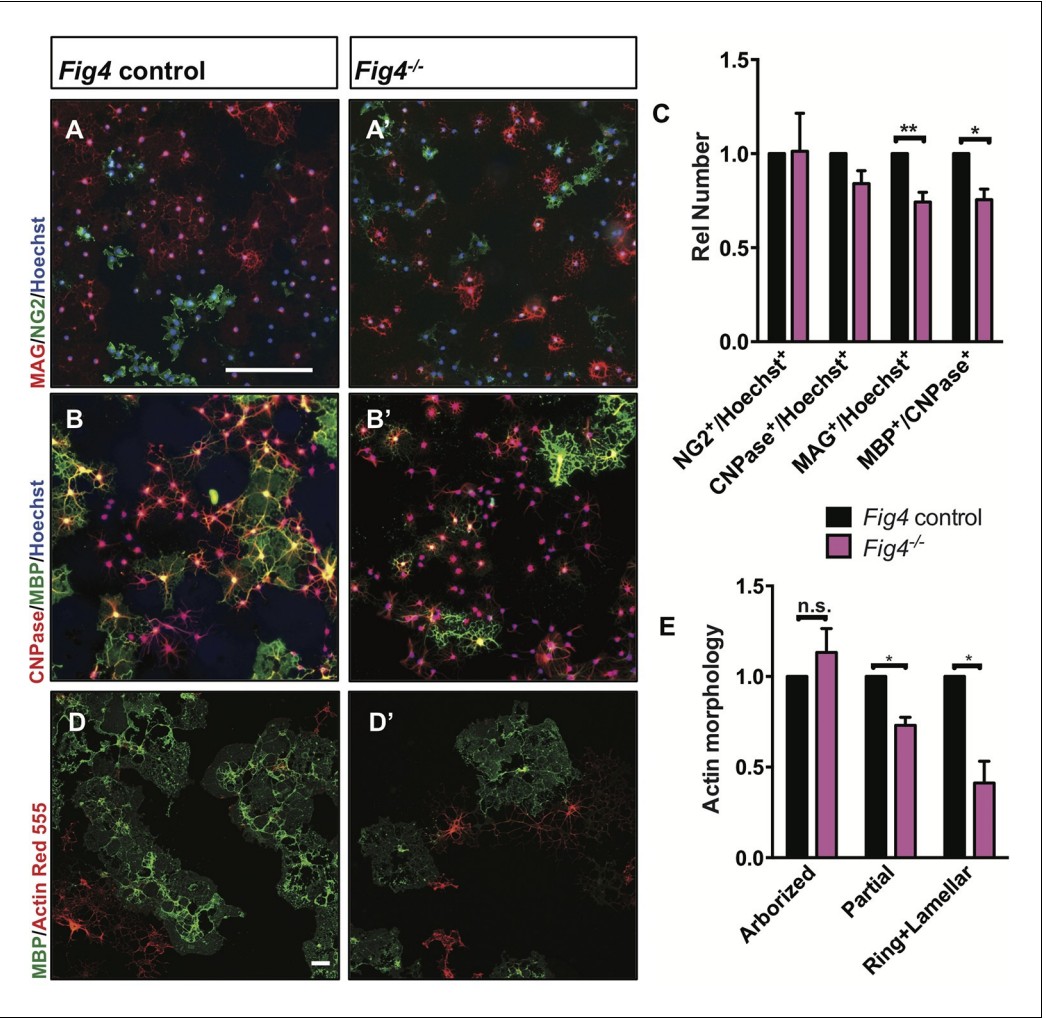

**Figure 5.** *Fig4*-deficient OLs show impaired differentiation and membrane expansion in vitro. Representative images of *Fig4* control (*Fig4*$^{+/+}$ *or Fig4*$^{+/-}$) and *Fig4*$^{-/-}$ primary OLs after 4 days in differentiation medium, fixed and stained for the OL-lineage markers (**A** and **A'**) NG2 and MAG; (**B** and **B'**) CNPase and MBP. Scale bar in **A-B'**, 200 µm. (**C**) Quantification of NG2, CNPase, MAG, and MBP/CNPase labeled cells in *Fig4* control (n = 3) and *Fig4*$^{-/-}$(n = 3) cultures normalized to Hoechst 33342 dye labeled cells. The ratio of immunolabeled cells over Hoechst$^{+}$ cells in *Fig4* control cultures was set at 1. Results are shown as mean value ± SEM, multiple t-test analysis with Holm-Sidak method. \*\*p=0.0075 (MAG), \*p=0.012 (MBP). (**D** and **D'**) Confocal images of MBP$^{+}$ and Actin Red 555$^{+}$ OLs in *Fig4* control and *Fig4*$^{-/-}$ cultures. Nuclei were labeled with TO-PRO-3, scale bar = 20 µm. (**E**) Quantification of the fraction of "arborized" (actin rich, no MBP), "partial" (partial actin disassembly, onset of MBP expansion), and "ring + lamellar" (full MBP expansion, actin disassembly) in *Fig4* control cultures (n = 4) and *Fig4*$^{-/-}$ (n = 4) cultures. Results are shown as mean value ± SEM, multiple t-test analysis with Holm-Sidak method. \*p=0.0008 ("partial"), \*p=0.009 ("ring + lamellar"). The effects of *Fig4* deletion on OPC proliferation and OL survival are shown in ***Figure 5—figure supplement 1***. Quantitative Western blot analysis of myelin proteins in primary OL lysates is shown in ***Figure 5—figure supplement 2***.

The following figure supplements are available for figure 5:

**Figure supplement 1.** Loss of *Fig4*$^{-/-}$ in primary OLs does not affect cell proliferation or cell death.

**Figure supplement 2.** Capillary Western analysis of primary OL lysates.

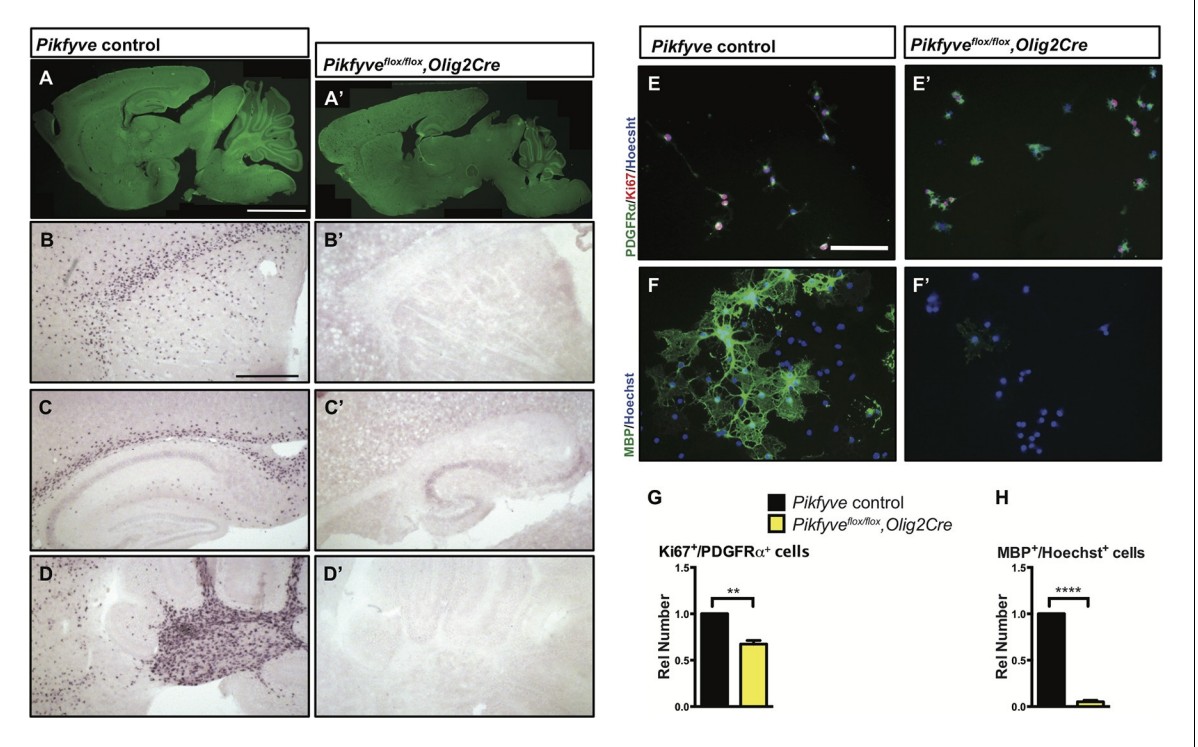

**Figure 6.** Conditional deletion of *Pikfyve* in OLs results in profound CNS hypomyelination. (**A-D'**) Sagittal sections of P13 mouse brains. (**A**) *Pikfyve* control (*Pikfyve*<sup>flox/+</sup> or *Pikfyve*<sup>flox/flox</sup>; n = 3) mice and (**A'**) *Pikfyve* conditional null (*Pikfyve*<sup>flox/flox</sup>,*Olig2Cre*; n = 3) mice stained with FluoroMyelin Green. In *Pikfyve*<sup>flox/flox</sup>,*Olig2Cre*, no myelin staining was observed, Scale bar, 1 mm. (**B-D'**) in situ hybridization for *Plp1* shows virtual absence of mature OLs in P13 *Pikfyve*<sup>flox/flox</sup>,*Olig2Cre* brain tissue, including (**B** and **B'**) internal capsule and corpus callosum, (**C'** and **C'**) hippocampus and corpus callosum and (**D** and **D'**) cerebellar white matter. Scale bar (**B-D'**) = 500 μm. (**E-H**) Cultures of primary OPCs/OLs isolated from *Pikfyve* control and *Pikfyve*<sup>flox/flox</sup>, *Olig2cre* mouse pups. (**E, E'**) At DIV2, cells were fixed and stained with anti-PDGFRα, anti-Ki67and Hoechst 33342 dye. (**F, F'**) After 3 days in differentiation medium, supplemented with T3, cells were fixed and stained with anti-MBP and Hoechst 33342. (**G**) Quantification of proliferating OPCs revealed a *Pikfyve*-dependent reduction in Ki67<sup>+</sup>/PDGFRα<sup>+</sup> double-labeled cells (n = 3 experiments per genotype). (**H**) Quantification of MBP<sup>+</sup> OLs normalized to Hoechst<sup>+</sup> cells shows a highly significant decrease in the number of MBP<sup>+</sup> OLs in *Pikfyve*<sup>flox/flox</sup>,*Olig2cre* cultures (n = 3 experiments per genotype). Unpaired Student's *t*-test; mean value ± SEM. **p=0.011 and ****p<0.0001. Toluidine blue labeling of epoxy resin embedded optics nerves of *Pikfyve* control and *Pikfyve*<sup>flox/flox</sup>,*Olig2cre* conditional mutant mice is shown in *Figure 6—figure supplement 1*.

The following figure supplement is available for figure 6:

**Figure supplement 1.** Optic nerve axons are not myelinated in *Pikfyve*<sup>flox/flox</sup>,*Olig2Cre* mice.

PIKFYVE, but not with FIG4 (*Figure 7A*). Similar to *Fig4*<sup>-/-</sup> mice, *Vac14*<sup>L156R/L156R</sup> mice exhibit ~50% reduction in PI(3,5)P$_2$. Immunoblots of brain membranes prepared from *Vac14*<sup>L156R/L156R</sup> mice showed significantly reduced levels of the myelin markers MAG, CNPase, and MBP (*Figure 7B–E*). The electrical properties of optic nerve from *Vac14*<sup>L156R</sup> homozygous mice were also impaired, with a significant increase in the population of slowly conducting fibers (*Figure 7F–H*). Consistent with this observation, toluidine blue staining of optic nerve sections of adult wild-type mice revealed many myelinated fibers but optic nerves of adult *Vac14*<sup>L156R/L156R</sup> mice showed few myelinated fibers (*Figure 7—figure supplement 1A–D*). Thus, independent genetic disruptions of the FIG4/PIKFYVE/VAC14 enzyme complex all result in severe hypomyelination and a PI(3,5)P$_2$ dosage-dependent decline in CNS white matter development.

## Myelin proteins are present within enlarged LAMP1<sup>+</sup> perinuclear vacuoles in primary OLs from *Fig4*<sup>-/-</sup> mice

The FIG4/PIKFYVE/VAC14 biosynthetic complex regulates intracellular PI(3,5)P$_2$ and thereby influences membrane trafficking through the endo-lysosomal system. DIV2 primary OPC cultures

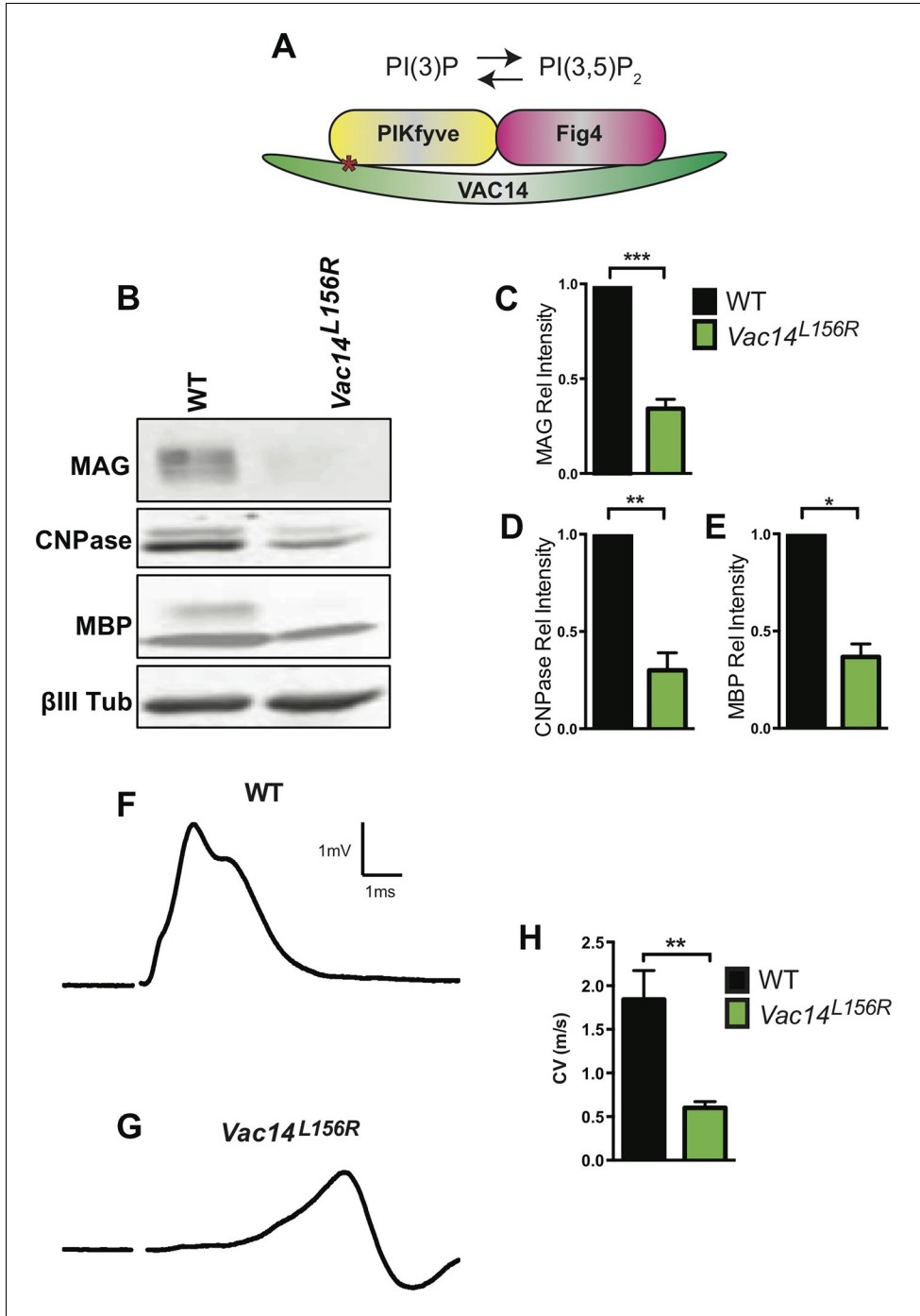

**Figure 7.** Homozygosity for *VAC14^{L156R}* leads to CNS hypomyelination and impaired conduction of compound action potentials. (**A**) Schematic of the PIKfyve/Vac14/Fig4 enzyme complex and its phosphoinositide products PI (3)P and PI(3,5)P$_2$. The red asterisk in VAC14 indicates the L156R point mutation that perturbs the interaction with PIKfyve, but not with Fig4. (**B**) Western blot analysis of brain membranes prepared from adult (P90-120) WT and *VAC14^{L156R}/VAC14^{L156R}* littermate mice revealed a reduction in the myelin markers MAG, CNPase, and MBP. Anti-class III β-tubulin (βIII-Tub), a neuronal marker, is shown as a loading control. (**C-E**) Quantification of protein bands detected by Western blotting, shows a significant decrease in MAG, CNPase, and MBP in *VAC14* mutant brain tissue (n = 3 independent blots per genotype). Unpaired Student's *t*-test; mean value ± SEM. ***p<0.001, **p=0.0015 and *p=0.0238. (**F** and **G**) Representative CAP traces recorded from acutely isolated optic nerves of WT and *VAC14^{L156T}* homozygous mice. (**H**) Quantification of average conduction velocity (CV) of largest amplitude peaks identified in **F** and **G**. Results are shown as mean value ± SEM, unpaired Student's *t*-test, **p=0.0063. WT
*Figure 7 continued on next page*

*Figure 7 continued*

n = 6 nerves, 3 mice and for *Vac14$^{L156R}$* mutants n = 6 nerves, 3 mice. Toluidine blue staining of epoxy resin embedded optic nerve sections from *VAC14$^{L156R}$/VAC14$^{L156R}$* mice is shown in *Figure 7—figure supplement 1*.
The following figure supplement is available for figure 7:

**Figure supplement 1.** Severe optic nerve hypomyelination in *VAC14$^{L156R/L156R}$* mice.

established from *Fig4* control (*Fig4$^{+/+}$* or *Fig4$^{-/+}$*) and *Fig4$^{-/-}$* mice were fixed and subjected to anti-LAMP1 and anti-PDGFRα double-immunofluorescence labeling. The majority of *Fig4$^{-/-}$* OPCs showed normal-sized lysosomes with a diameter of < 1 μm, while a fewcells (< 20% ) exhibited enlarged LAMP1$^+$ vesicles (*Figure 8—figure supplement 1A–B"*). Upon OL differentiation, an increase in size and number of perinuclear LAMP1$^+$ vesicles is observed in *Fig4$^{-/-}$*cultures. The enlarged perinuclear LAMP1$^+$ structures are prominently labeled with anti-MAG (*Figure 8—figure supplement 1C–D"*). In a parallel approach, *Fig4$^{-/-}$* OLs were transfected with *Rab7-YFP*, a reporter for LE. Enlarged perinuclear vacuoles in Fig4$^{-/-}$ OLs are positive for Rab7-YFP (*Figure 8—figure supplement 2A–A'*). Live imaging of primary OLs revealed that the majority of enlarged perinuclear vacuoles in Fig4$^{-/-}$ OLs are stable for several days. However, vacuole size varies and live imaging revealed that some vacuoles appear and disappear over a period of 12 hr (*Video 5*). Collectively, these studies demonstrate that in *Fig4$^{-/-}$* OLs, myelin building blocks that are normally trafficked through the LE/Lys are present in abnormal, enlarged vesicles the majority of which is stable for several days.

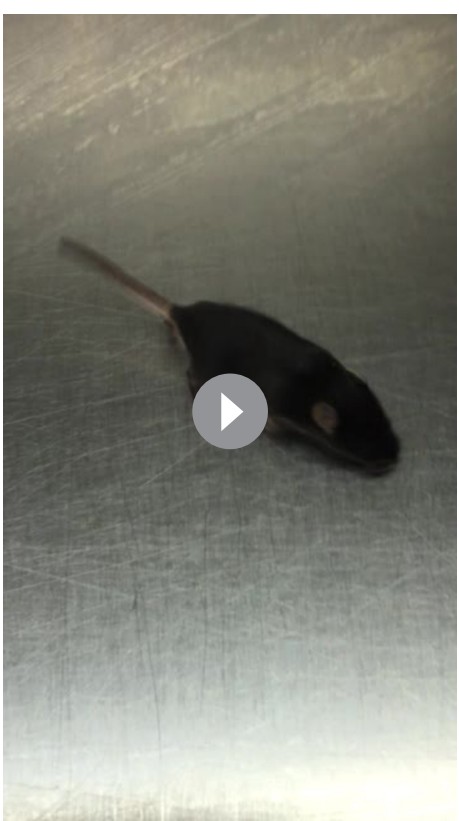

**Video 3.** Normal locomotion of juvenile *Pikfyve$^{flox/+}$* control mice. A representative video of a control *Pikfyve$^{flox/+}$* mouse at P13. N = 16

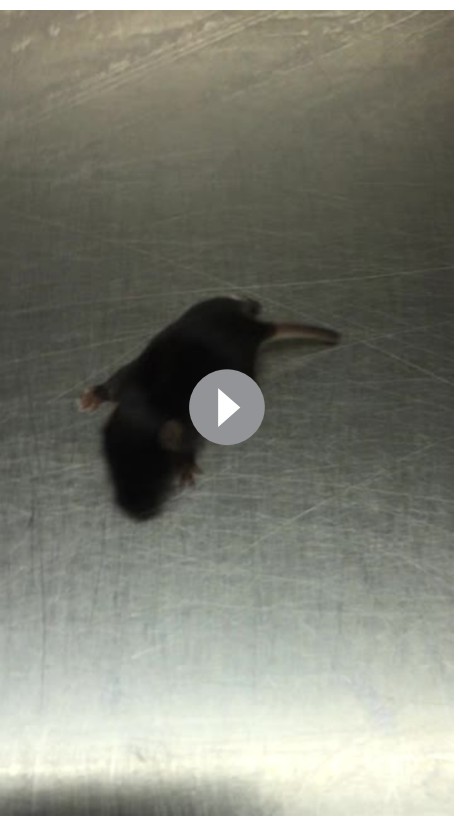

**Video 4.** Severe tremor in juvenile *Pikfyve$^{flox/flox}$*, *Olig2Cre* mice. A representative video of a *Pikfyve$^{flox/flox}$*,*Olig2Cre* conditional mutant mouse at P13 reveals a severe tremor phenotype. N = 16

## Cell surface derived MAG is trapped in large vacuoles in the LE/Lys compartment in *Fig4*[-/-] OLs

In developing OLs, myelin proteins such as MAG and PLP transiently accumulate on the plasma membrane (PM) at the cell soma, prior to undergoing endocytosis and LE/Lys dependent transport to the myelin sheet (*Winterstein et al., 2008*). To monitor trafficking of MAG, we used antibody tagging in live OL cultures. In wildtype OLs, anti-MAG-Alexa488 binds to MAG on the PM surface, undergoes endocytosis and is targeted to LAMP1[+] vesicles in the LE/Lys compartment (*Figure 8—figure supplement 2B–B"*). In these wildtype cultures, anti-MAG[+] vesicles are small, with a median volume of $0.3 \pm 0.06$ μm$^3$, and partially overlap with LysoTracker[+] vesicles (*Figure 8A–A"*). In contrast, in *Fig4*[-/-] OLs, anti-MAG-Alexa488 is endocytosed and accumulates in LAMP1[+] perinuclear vacuoles with greatly enlarged size ($\geq 5$ μm$^3$, mean volume $94 \pm 41$ μm$^3$) and also in smaller MAG[+]/LAMP1[+] vesicles with a median volume of $0.7 \pm 0.25$ μm$^3$. The average size of all vesicles in *Fig4*[-/-] OLs is $1.65 \pm 0.32$ μm$^3$ (*Figure 8B–B" and C*, *Figure 8—figure supplement 2C–C"*). This suggests that independent of *Fig4* genotype, MAG is transported to the PM and is rapidly endocytosed. In *Fig4*[-/-] OLs, large MAG[+]/LAMP1[+] vesicles rarely overlap with LysoTracker staining (*Figure 8B-8B"*), suggesting that large vesicles may exhibit reduced acidification. As an independent approach to assess whether perturbation of PI(3,5)P$_2$ synthesis causes accumulation of MAG in large perinuclear vacuoles, wildtype OL cultures were treated with 1 μM apilimod, a potent inhibitor of PIKfyve (*Cai et al., 2013*). Treatment with apilimod for 90–120 min leads to the formation of large perinuclear vacuoles laden with MAG (*Figure 8D–D"*), similar to those in *Fig4*[-/-] OLs. To evaluate the specificity of the anti-MAG-Alexa488 antibody, experiments were repeated with primary OLs isolated from *Mag*[-/-] pups (*Pan et al., 2005*). Bath application of anti-MAG-Alexa488 to *Mag*[-/-] OLs treated with vehicle or apilimod did not result in immunostaining, demonstrating that the antibody is specific for MAG (*Figure 8—figure supplement 3A–D"*). The myelin protein MOG has a different endocytotic fate from MAG, trafficking through recycling endosomes (RE) but not the lysosomal compartment (*Winterstein et al., 2008*). Simultaneous antibody labeling of cell surface MAG and MOG in live OLs confirmed distinct endocytotic trafficking routes in both *Fig4* control and *Fig4*[-/-] cultures. Importantly, in *Fig4*[-/-] OLs, MOG was not present in the enlarged vacuoles that are typically laden with MAG (*Figure 8—figure supplement 4A–B"*). This suggests that the defect in *Fig4*[-/-] OLs in trafficking of myelin building blocks from the PM is specific for trafficking through the LE/Lys compartment and does not affect trafficking through the RE.

## *Fig4*[-/-] OLs display impaired MAG trafficking through the LE/Lys compartment

The perinuclear location and large size of MAG[+] vacuoles suggests that their mobility may be compromised, potentially leading to impaired trafficking of MAG and other myelin building blocks transported via the LE/Lys route. To explore this possibility, we assessed movement of MAG[+] vesicles in live OLs using time-lapse imaging (*Figure 9A–B'*). Small vesicles labeled with anti-MAG-Alexa488 are observed in *Fig4*[+/+] and *Fig4*[-/-] primary OLs, with average volumes of 0.3 μm$^3$ and 0.7 μm$^3$, respectively. The average velocity of these 'normal-sized' vesicles is comparable in *Fig4*[+/+] and *Fig4*[-/-] cells: $0.09 \pm 0.01$ μm/s and $0.07 \pm 0.01$ μm/s, respectively (*Figure 9C*). The large MAG[+] vesicles in the *Fig4*[-/-] OLs with an average volume of $94 \pm 41$ μm$^3$ are more stationery, with an average velocity of $0.033 \pm 0.005$ μm/s (*Figure 9C*), and they fail to reach the nascent myelin sheet. These data suggest that trafficking of MAG and other LE/Lys dependent myelin building blocks is impaired in the *Fig4*[-/-]

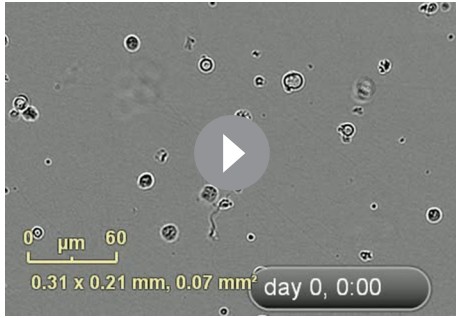

**Video 5.** Vacuoles in *Fig4*[-/-] OLs appear and disappear within hours. Time-lapse live cell analysis of *Fig4*[-/-] primary OPC/OLs imaged with an IncuCyte ZOOM microscope. Phase contrast images were taken every 2 hr over a time interval of 60 hr. The majority of *Fig4*[-/-] cells contain large perinuclear vacuoles. Some of these vacuoles appear and disappear within hours (n = 3). Scale bar = 60 μm.

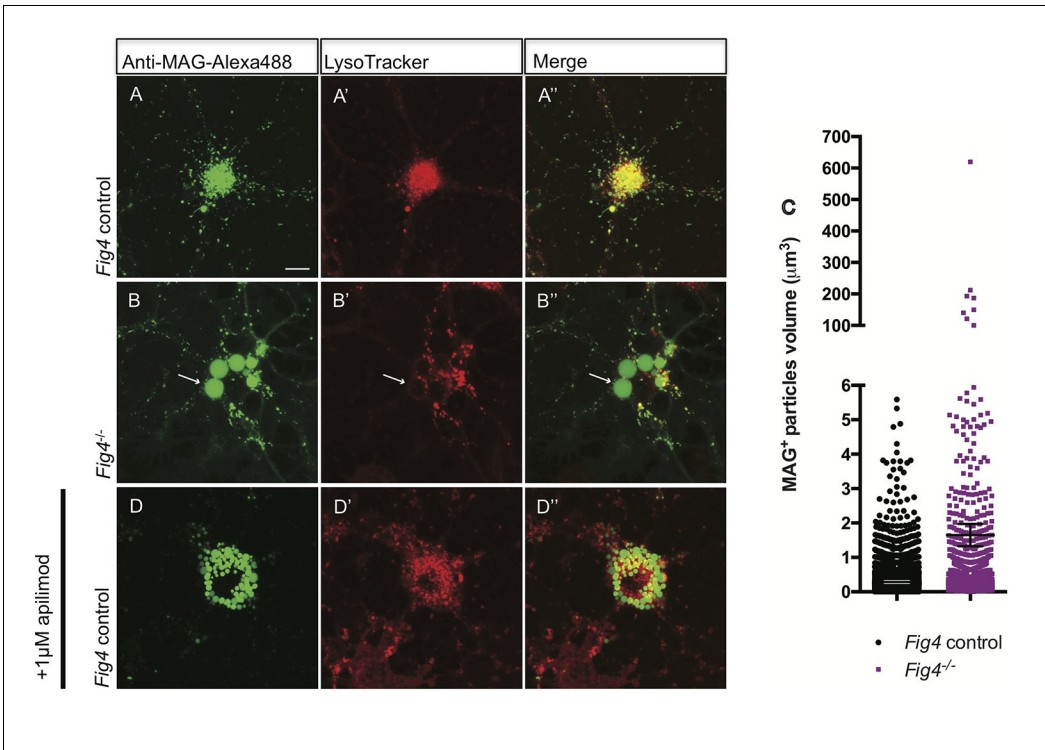

**Figure 8.** In *Fig4⁻ᐟ⁻* OLs, MAG accumulates in large perinuclear vacuoles. Confocal images of live OLs acutely labeled with bath applied anti-MAG-Alexa488 (green) and LysoTracker Deep Red. (**A-A''**) *Fig4* control (*Fig4⁺ᐟ⁺* or *Fig4⁺ᐟ⁻*) OLs incubated with anti-MAG-Alexa488 and LysoTracker, single channel and merged images are shown. (**B-B''**) *Fig4⁻ᐟ⁻* OLs incubated with anti-MAG-Alexa488 and Lysotracker shows accumulation of MAG in large perinuclear vacuoles (arrows), single channel and merged images are shown. Of note, large perinuclar MAG⁺ vacuoles do not stain with LysoTracker. (**C**) Scatter plot depicting the volume of anti-MAG-Alexa488⁺ particles in live *Fig4* control and *Fig4⁻ᐟ⁻* OLs. Each dot represents an individual vesicle (n = 4 experiments, 9 cells per genotype). Mean volumes ± SEM are shown. (**D-D''**) Wildtype OLs were incubated with anti-MAG-Alexa488 and LysoTracker and then acutely treated with the PIKfyve inhibitor apilimod. MAG accumulates in large perinuclear vacuoles, the majority of which does not stain with LysoTracker (n = 4 for *Fig4* controls and n = 4 for *Fig4⁻ᐟ⁻*cultures). For apilimod treatment, n = 3 independent cultures. Maximum projection confocal z-stack images are shown, scale bar = 10 μm. Further characterization of enlarged perinuclear vacuoles in *Fig4⁻ᐟ⁻* OL cultures, specificity control for the anti-MAG-Alexa488 antibody and distinct trafficking routes of MAG and MOG are shown in *Figure 8—figure supplement 1–3* and *4*.

The following figure supplements are available for figure 8:

**Figure supplement 1.** *Fig4⁻ᐟ⁻* OLs show enlarged perinuclear vacuoles that stain positive for LAMP1.

**Figure supplement 2.** In *Fig4⁻ᐟ⁻* OLs, PM derived MAG is transported to enlarged vesicles in the LE/Lys compartment.

**Figure supplement 3.** Specificity control for anti-MAG-Alexa488 antibody.

**Figure supplement 4.** Live imaging of primary OLs reveals distinct trafficking routes for PM-derived MAG and MOG.

OLs. Collectively, these studies indicate that PI(3,5)P$_2$ is critical for myelin protein trafficking through the LE/Lys compartment in developing OLs.

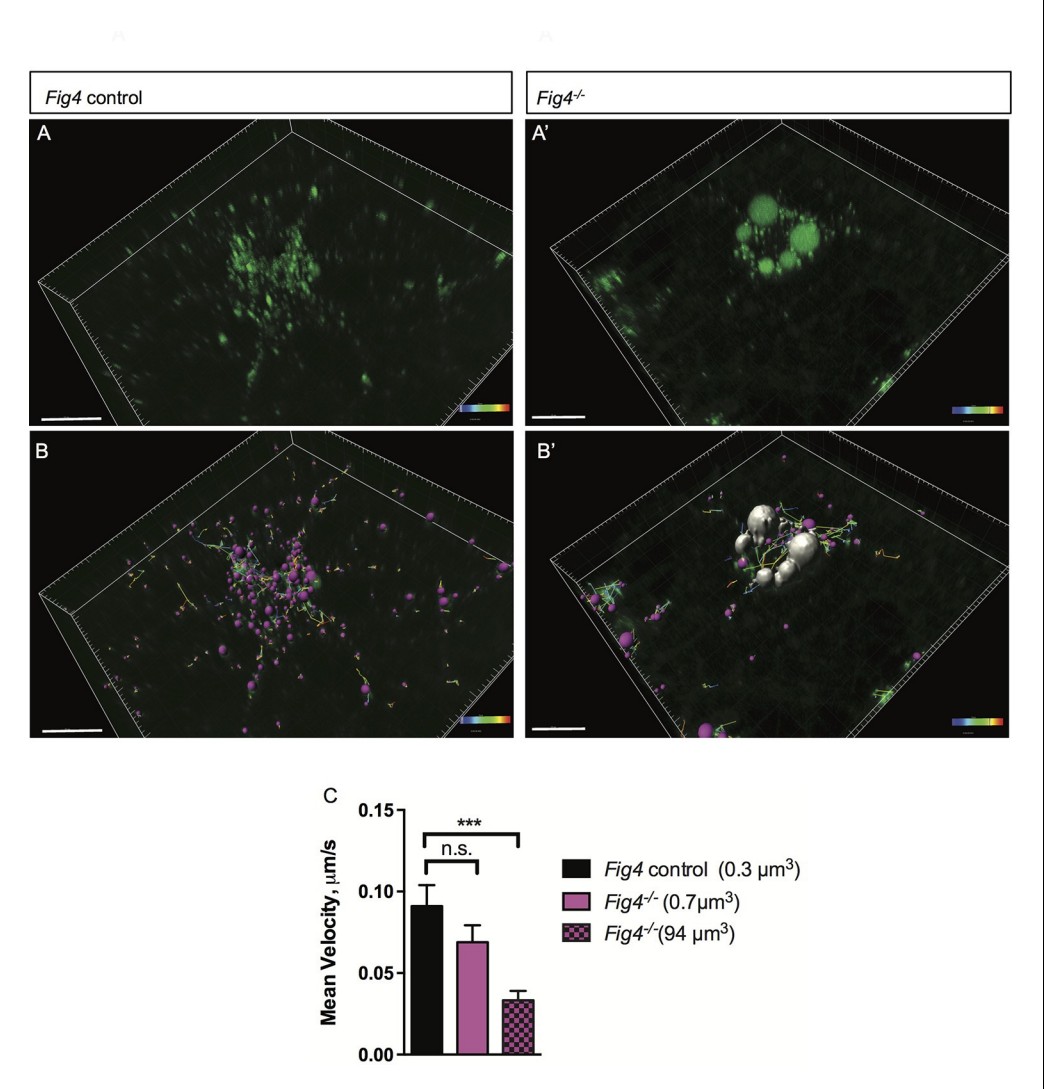

**Figure 9.** In *Fig4*<sup>-/-</sup> OLs, vesicular trafficking through the LE/Lys compartment is defective. Representative confocal images of live, anti-MAG-Alex488 labeled (A) *Fig4* control OLs and (A') *Fig4*<sup>-/-</sup> OLs. Time-lapse imaging was used to track movement of MAG$^+$ vesicles. (B) Using Imaris software, MAG$^+$ vesicles were labeled with pink spheres and vesicular movement was tracked (yellow lines) in *Fig4* control cultures. (B') Imaris software was used to track movement of large vesicles (white color) and small vesicles (purple color) in *Fig4*<sup>-/-</sup> OLs: tracks of individual vesicles are shown. (C) Quantification of mean velocity of MAG$^+$ vesicles in *Fig4* control OLs and *Fig4*<sup>-/-</sup> OLs. In *Fig4*<sup>-/-</sup> OLs, movement of small vesicles (0.7 $\mu m^3$) and large vesicles (94 $\mu m^3$) was assessed separately. The velocity is shown as mean value ± SEM. N = 4 independent experiments and a total of 9 cells per genotype were analyzed. One-way ANOVA with Dunnett posthoc, ***p= 0.001. (n.s. = not significant).

## PI(3,5)P$_2$ is important for myelin membrane trafficking in live brain slices

Inter-cellular communication is critical for proper development of the axo-glial unit. To extend the studies of myelin protein trafficking to a system that contains intact axo-glial units, we prepared acute forebrain slices from P10-P14 mice and kept them in oxygenated artificial cerebrospinal fluid. Trafficking of MAG was monitored by bath application of mouse anti-MAG-Alexa555 for 2 hr at 32°C. To distinguish between endocytosed MAG and PM localized MAG, brain slices were fixed and incubated with a secondary anti-mouse-Alexa488 conjugated antibody under non-permeabilizing conditions. Endocytosed MAG containing vesicles were prominently found in OL perinuclear regions

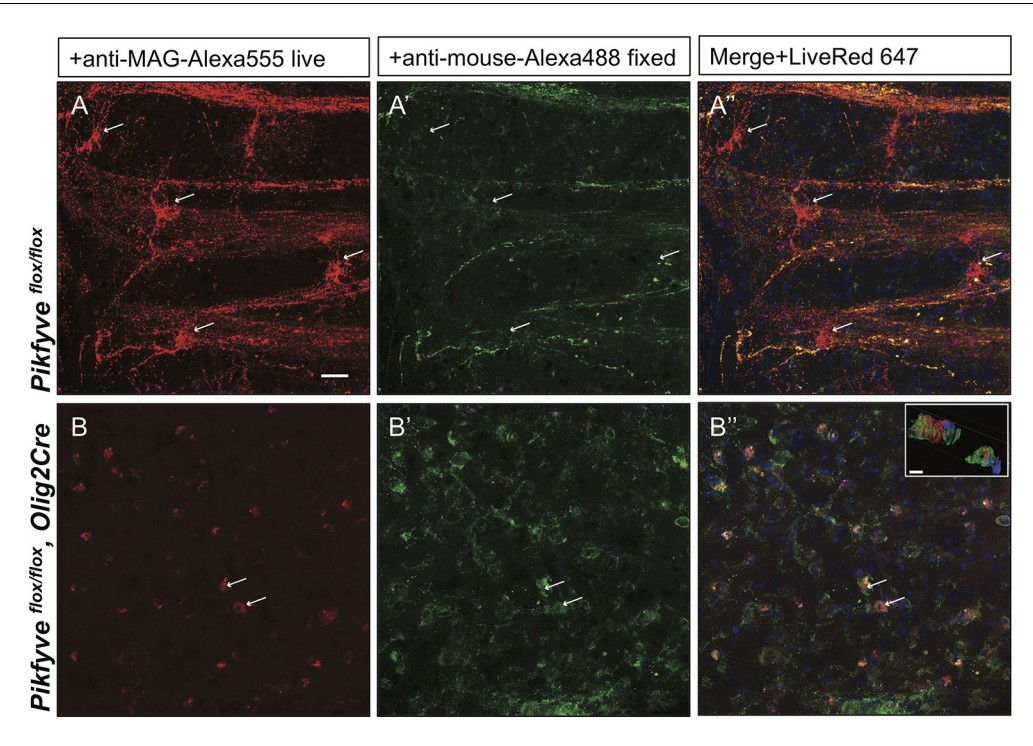

**Figure 10.** Impaired trafficking of MAG in *Pikfyve*$^{flox/flox}$,*Olig2Cre* brain slices. Confocal images of acute brain slices in oxygenated ACSF treated with bath-applied anti-MAG-Alexa555 antibody, fixed and stained with anti-mouse-Alexa488 secondary antibody to distinguish between endocytosed MAG (red) and PM localized MAG (green). (**A**) OLs in the striatum of *Pikfyve* control mice (P13) show punctate MAG labeling in the cell soma (arrows) and along processes that form internodes. Only few MAG$^+$ structures are also stained with anti-mouse-Alexa488, and thus, localized on the PM. (**B-B″**) Limited perinuclear MAG labeling is observed in the *Pikfyve*$^{flox/flox}$,*Olig2Cre* striatum. Many MAG$^+$ structures are labeled red and green, and thus localized to the PM, however intracellular MAG is observed in some cells. Scale bar = 20 μm. Small inset shows a 3D view of the two cells labeled with arrows (**B-B″**). MAG$^+$ vesicles (red) only partially overlap with PM localized MAG (green). Alexa488$^+$ isosurface transparency is adjusted to 50% to demonstrate intracellular Alexa555$^+$ (red) and LiveNuc 647$^+$ (blue) structures. Scale bar = 10 μm. To directly demonstrate that anti-MAG antibody labeled cells are OLs, parallel experiments were carried out with brain slices of *LacZ/EGFP,Olig2Cre* reporter mice. Anti-MAG antibody specificity is demonstrated with *Mag*$^{-/-}$slices in *Figure 10—figure supplement 1*.

The following figure supplement is available for figure 10:

**Figure supplement 1.** Anti-MAG labeling of EGFP$^+$ OLs and specificity control for anti-MAG antibody in acute brain slices.

and along cellular processes that form the myelin internode (*Figure 10A–A″*). Only a small fraction of MAG is labeled with both antibodies, and thus localized to the PM on the cell surface (*Figure 10A–A″*). To visualize cells in the OL lineage, we repeated MAG trafficking studies with brain slices from the *ROSA-LacZ/EGFP,Olig2Cre* reporter mouse. Vesicular MAG labeling was abundant in EGFP$^+$ cells, indicating that endocytosis of PM localized MAG does occur in cells of the OL-lineage and vesicular labeling is not the result of nonspecific antibody uptake by microglia or other cell types (*Figure 10—figure supplement 1A–C*). To control for antibody specificity, brain slices from *Mag*$^{-/-}$ mice were processed in parallel and revealed no significant labeling (*Figure 10—figure supplement 1D–F*). Thus, acute brain slices provide an opportunity to study myelin protein trafficking in live tissue. To assess whether PI(3,5)P$_2$ is required for endocytosis and trafficking of PM derived MAG in live brain tissue, the experiment was repeated with forebrain slices prepared from *Pikfyve*$^{flox/flox}$, *Olig2Cre* pups. Strikingly, in the absence of PI(3,5)P$_2$, MAG$^+$ labeling was restricted to abnormal perinuclear accumulations, and trafficking to cell processes was virtually absent (*Figure 10B′–B″*).

The data demonstrate that in brain slices, as well as cultured cells, PI(3,5)P$_2$ is required for proper membrane trafficking from the PM through the LE/Lys compartment.

## Discussion

Multiple independent means of perturbing the Fig4/Vac14/PIKfyve enzyme complex all lead to profound CNS hypomyelination. Remarkably, conditional ablation of *Fig4* either in neurons or OLs is sufficient to disrupt normal CNS myelination, indicating that both OL-intrinsic and OL-extrinsic mechanisms of OL maturation function in a *Fig4*-dependent manner. The hypomyelination phenotype in *Fig4* conditional mutants and *VAC14$^{L156R}$* mice is physiologically relevant since it is associated with substantially reduced amplitude and conduction velocity of compound action potentials. Primary OPCs deficient in *Fig4* progress normally to the stage of NFOs but their differentiation into mature OLs is impaired. In *Fig4$^{-/-}$* OLs, MAG is trafficked to the PM, undergoes endocytosis and is localized to enlarged LAMP1$^+$ perinuclear vacuoles. The reduced motility of the enlarged MAG/LAMP1$^+$ vacuoles and their perinuclear position suggests that myelin building blocks are trapped in the LE/Lys compartment and cannot be delivered to the developing myelin sheath. Conditional deletion of *Pikfyve* in the OL lineage leads to more pronounced defects characterized by impaired OL differentiation, greatly reduced myelin membrane trafficking and profound CNS dysmyelination. Together, these studies firmly establish a critical role for the FIG4/PIKFYVE/VAC14 enzyme complex, and by extension its lipid product PI(3,5)P$_2$, in myelin protein trafficking through the LE/Lys system in developing OLs and proper assembly of the axo-glial unit.

### Impaired PI(3,5)P$_2$ metabolism attenuates OL differentiation

Immunohistological studies of *Fig4$^{-/flox}$,Olig2Cre* optic nerves and experiments with *Fig4$^{-/-}$* primary OLs did not detect a significant change in OPC density or reduction in viability. OPCs deficient for *Fig4* progress and differentiate normally to the stage of newly formed OLs (NFOs), a postmitotic cell type characterized as PDGFRα$^-$, GalC$^+$, MOG$^-$ (*Zhang et al., 2014*). However, differentiation of NFOs into mature OLs is PI(3,5)P$_2$-dosage dependent. The arrest of OL differentiation becomes more severe as PI(3,5)P$_2$ levels are reduced to ~50% of wildtype levels, in *Fig4* and *VAC14* mice, or completely depleted in *Pikfyve* mutant mice. OL maturation is highly regulated, and can be attenuated or blocked by perturbation of numerous signaling pathways and transcriptional programs (*Emery et al., 2009*; *Bercury and Macklin, 2015*; *Marinelli et al., 2016*). The fate of immature OLs that fail to progress to the mature stage remains unclear. However, these cells are likely to be short-lived and destined to die. The number of activated caspase-3$^+$ cells in the OL lineage of *Fig4$^{-/-}$* mice is not significantly increased (*Winters et al., 2011*), suggesting that immature OLs either do not die in large numbers or die in a caspase-independent manner. Additional studies are needed to determine exactly at which stage of OL lineage progression PI(3,5)P$_2$ deficiency impairs differentiation and how PI(3,5)P$_2$ regulates progression to a mature myelin producing cell.

### *Fig4*-dependent trafficking of myelin building blocks through the LE/Lys

Like epithelial cells, OLs are polarized, with the myelin sheath resembling the apical membrane domain and the membrane near the OL cell body the basolateral membrane domain (*Salzer, 2003*; *Maier et al., 2008*; *Masaki, 2012*). Myelin-producing OLs synthesize and transport large quantities of myelin building blocks (lipids and proteins) in order to segmentally ensheath multiple axons. Myelinogenesis also requires membrane sorting and trafficking to specific subdomains of the nascent myelin membrane sheath. Indeed, the final destination of myelin proteins may vary between compact myelin (e.g. PLP), peri-axonal loops (MAG) or abaxonal loops (MOG) of non-compact myelin (*Arroyo and Scherer, 2000*, *Salzer, 2003*; *White and Kramer-Albers, 2014*). As in other polarized cells, OL proteins may be targeted through direct transport pathways from the Golgi to their final destination (*Salzer, 2003*). Alternative strategies are also employed to target key myelin constituents to their final destination. The mRNA for MBP, encoding a protein important for axon wrapping and myelin compaction, is packaged into RNA-granules and transported to distal sites within OL processes for regulated translation (*Müller et al., 2013*). MAG, PLP, and MOG are synthesized in the endoplasmic reticulum and transported through the Golgi network to the PM near the OL cell body (analogous to the basolateral domain) as an intermediate target. From there MOG is targeted to the recycling endosome (RE)

while MAG and PLP are targeted to the LE/Lys for delivery to the myelin sheath (analogous to the apical membrane domain) (*Simons and Trajkovic, 2006*; *Maier et al., 2008*; *Winterstein et al., 2008*). LAMP1 is a marker for LE/Lys and we show that MAG is targeted to LAMP1$^+$ vesicles in both *Fig4$^{+/+}$* and *Fig4$^{-/-}$* OLs. A key feature of the MAG/LAMP1 double-labeled vesicles in *Fig4$^{-/-}$* mutant OLs is their greatly enlarged size and perinuclear position. The average velocity of these vesicles is significantly reduced, suggesting impaired membrane trafficking through the LE/Lys compartment. Trafficking defects in *Fig4$^{-/-}$* OLs are confined to the LE/Lys compartment as trafficking of MOG through RE occurs apparently normal, independent of *Fig4* genotype. The severe CNS hypomyelination phenotype in *Fig4$^{-/flox}$,Olig2Cre* mice is likely not the result of impaired MAG trafficking alone, but rather the result of mistrafficking of numerous myelin building blocks normally migrating through the LE/Lys compartment. For example, cholesterol (in part bound to PLP) and glycosphingolipids are endocytosed from the PM and stored in LE/Lys vesicles (*Trajkovic et al., 2006*, *Winterstein et al., 2008*). During OL maturation, neuronal signals trigger a profound redistribution of PLP-containing membrane domains; endocytosis is reduced and PLP together with cholesterol and glycosphingolipids is moved from the LE/Lys to the PM (*Trajkovic et al., 2006*). In humans, impaired trafficking of PLP due to mutation or altered dosage of the *Plp1* gene, causes Pelizaeus-Merzbacher disease (PMD) and Spastic Paraplegia Type 2 (SPG2), developmental disorders with severe neurological impairment (*Inoue, 2005*). Overexpression of PLP in mice leads to accumulation of the protein in autophagic vesicles and LE/Lys, leading to reduction of other myelin proteins such MBP, MAG, and MOG (*Karim et al., 2007*). As in *Fig4$^{-/-}$* mice, PMD results in reduced number of OLs and CNS dysmyelination. Failure of lysosomal trafficking or function is thus a common underlying mechanism for a growing number of hereditary disorders that cause CNS dysmyelination, including PMD, Niemann-Pick type C disease, and several lysosomal storage diseases (*Folkerth, 1999*; *Yaghootfam et al., 2005*; *Prolo et al., 2009*; *Schweitzer et al., 2009*; *Faust et al., 2010*; *Grishchuk et al., 2014*).

## PI(3,5)P$_2$-dependent trafficking of myelin membrane components in developing OLs

Different phosphoinositides exhibit unique distribution to intracellular membrane compartments and have been implicated as key regulators of membrane sorting and targeted vesicular trafficking (*Mayinger, 2012*). PI(3,5)P$_2$, for example, decorates vesicles in the LE/Lys compartment and serves as a docking site for cytosolic proteins (*Mayinger, 2012*). PIP binding proteins frequently interact with small GTPases belonging to the Rab or Arf families, establishing a combinatorial code that defines membrane identity (*Behnia and Munro, 2005*; *Stenmark, 2009*; *Jean and Kiger, 2012*; *Mayinger, 2012*; *Egami et al., 2014*). The phosphorylation status of PIPs and the activation state of small GTPases can be rapidly modified, providing an identification code that is both unique and dynamic, two prerequisites for targeted membrane transport. In HeLa cells, for example, the lysosomal membrane is characterized by the presence of PI(3,5)P$_2$ and the small GTPases Rab7 and Arf-like (*Bucci et al., 2000*; *Hofmann and Munro, 2006*). In fibroblasts cultured from *Fig4$^{-/-}$* or *VAC14$^{L156R/L156R}$* mice, PI(3,5)P$_2$ levels are reduced by ~50% leading to formation of greatly enlarged LAMP1$^+$ vacuoles (*Chow et al., 2007*; *Jin et al., 2008*; *Zou et al., 2015*). In *Fig4$^{-/-}$* OLs, Rab7-YFP localizes to large perinuclear vacuoles (*Figure 8—figure supplement 2*). In HeLa cells, overexpression of constitutively active Rab7 leads to formation of large LAMP1$^+$ and LAMP2$^+$ vacuoles (*Bucci et al., 2000*). A direct interaction of VAC14 with the Rab7 GTPase activating protein (GAP) TBC1D15 has recently been described in HeLa cells (*Schulze et al., 2014*). This suggests the existence of a large protein complex that controls the interconversion of PI(3)P and PI(3,5)P$_2$ and the activity of select Rab GTPases, an emerging theme for directed membrane trafficking (*Jean et al., 2015*). Rab GTPases constitute a large protein family whose members are localized to distinct intracellular membrane microdomains to coordinate vesicle trafficking (*Stenmark, 2009*; *Hutagalung and Novick, 2011*). The GTPase Rab3A is expressed in OLs and has been shown to participate in membrane trafficking and myelination (*Schardt et al., 2009*). As discussed above, transport of myelin membrane components, including PLP, cholesterol and MAG, involves membrane sorting and trafficking through the LE/Lys compartment prior to insertion into the nascent myelin sheath (*White and Kramer-Albers, 2014*). Thus, interference with PI(3,5)P$_2$ synthesis, turnover, or binding partners that define LE/Lys membrane identity results in impaired cargo delivery of key myelin membrane components required for membrane expansion and sheath formation.

## Neuronal *Fig4* participates in CNS myelination

The severe hypomyelination phenotype in *Fig4-/flox*, *SynCre* mice suggests that *Fig4*-dependent neuronal signals are necessary for proper CNS myelination. When coupled with our previous finding that transgenic *Fig4* directed by the NSE promoter on a *Fig4-/-*background (*Fig4-/-*,*NSE-Fig4*) rescues the myelination defect (*Winters et al., 2011*; *Ferguson et al., 2012*), this suggests that normal levels of *Fig4* in neurons is necessary for CNS myelination and that neuronal overexpression of recombinant *Fig4* on a global *Fig4-/-* background is sufficient to drive CNS myelination. Multiple lines of evidence have demonstrated that neuron-derived signals regulate OL maturation and axon myelination (*Coman et al., 2005*; *Trajkovic et al., 2006*; *Ohno et al., 2009*; *Winters et al., 2011*; *Yu and Lieberman, 2013*; *Yao et al., 2014*). We speculate that neuronal *Fig4* regulates LE/Lys-dependent transport and axonal presentation of a 'pro-myelination' signal(s) necessary for OL differentiation and CNS axon myelination and that transgenic overexpression of *Fig4* in neurons (NSE-Fig4) leads to an elevated production of 'pro-myelination' signals(s) sufficient to rescue the deficiency of *Fig4* in the OL lineage of the *Fig4-/-*,*NSE-Fig4* transgenic mice. Alternatively, neuronal *Fig4* may accelerate the loss of 'anti-myelination' signal(s) on the axonal surface, e.g., through endocytosis. Inter-cellular communication may occur through paracrine action of secreted molecules or shedding vesicles. Exosomes are extracellular vesicles produced by many cells that facilitate transport and exchange of proteins, mRNAs and regulatory RNAs with important functions in cellular processes including myelination (*Frühbeis et al., 2012*; *Pusic and Kraig, 2014*). Because *Fig4* plays an important role in membrane trafficking through the LE/Lys system, it is possible that protein secretion or the content and abundance of exosomes may be altered in the mutant mice. Two independent approaches to delete *Fig4* in the OL lineage (*Olig2Cre* and *PdgfαCreER*) revealed that *Fig4* is required in the OL lineage for proper CNS myelination. These data were corroborated by in vitro studies with primary OLs. Taken together, our observations suggest that endogenous levels of *Fig4* gene expression in both neurons and OLs are necessary for normal CNS myelination.

Technical limitations in the specificity of transgene promoters may affect the interpretation of these experiments. For neuron-specific loss-of-function we employed female *SynapsinCre/+* mice driven by a synapsin-1 gene (SYN1) promoter fragment (*Rempe et al., 2006*), and for neuron-specific gain-of-function studies we used a 4.6 kb *NSE* promoter fragment (*Winters et al., 2011*; *Ferguson et al., 2012*). While these are commonly used strategies, it is recognized that in the developing mouse the *NSE* (ENO2) and *SYN1* promoters may have some leakiness that results in transient expression in non-neuronal cells including glia. A low level of expression of the endogenous *SYN1* and *ENO2* genes in OPCs/OLs has been reported (*Zhang et al., 2014*), but it is not clear whether this expression is retained by the promoter fragments that were used to drive transgene expression. Independent of these technical limitations, we provide multiple lines of evidence that genetic manipulations that compromise PI(3,5)P$_2$ synthesis profoundly impact OL differentiation and CNS myelination.

## Novel assay to monitor myelin protein trafficking in brain tissue

Acutely prepared brain slices are viable for several hours when maintained in oxygenated ACSF, a method commonly used for electrophysiological recordings (*Lee et al., 2008*). Studies with primary OLs suggest that newly synthesized myelin proteins are initially transported to the PM near the cell soma where they interact with lipids and other myelin proteins (*Winterstein et al., 2008*). These myelin-like structures are then thought to be endocytosed and trafficked to specific subdomains of the nascent myelin membrane sheath. Using acute brain slices combined with genetic labeling of cell in the OL lineage and confocal microscopy, we show that antibody-labeled MAG on the PM becomes rapidly endocytosed and is found in small vesicles in the OL cell soma and long processes that form internodes. Since sorting and trafficking of myelin building blocks are key components of myelinogenesis, future studies using acute brain slices may be productively combined with pharmacological and genetic manipulations to obtain detailed understanding of membrane trafficking in developing OLs.

## Materials and methods

### Transgenic mice

All mice were housed and cared for in accordance with NIH guidelines, and all research conducted was done with the approval of the University of Michigan Committee on Use and Care of Animals. The spontaneous $Fig4^{-/-}$ null mutation plt (*Chow et al., 2007*) is maintained as two congenic lines, C57BL/6J.plt/+ and C3HeB/FeJ.plt/+. F1 plt/plt homozygotes obtained from crosses between these lines survive to 30–45 days, permitting analysis of myelination, and these were used for most experiments. A subset of in vitro experiments was carried out on cells from the C3HeB/FeJ.plt congenic mice. The conditional $Fig4^{flox}$ allele was described elsewhere (*Ferguson et al., 2012*) and is maintained on strain C3HeB/FeJ from which the retinal degeneration locus *rd* was removed by repeated backcrossing and selection. Neuron-specific conditional knockout mice ($Fig4^{-/flox}$,*SynCre*) were generated and maintained as previously described (*Ferguson et al., 2012*). The *Olig2Cre/+* line (*Schüller et al., 2008*) and the *PdgfraCre-ER/+* (*Kang et al., 2010*) (Jackson Laboratory stock # 018280) were used to delete *Fig4* in the OL lineage. For inducible gene ablation in $Fig4^{-/flox}$, *PdfrαCreER* mice, 4-hydroxytamoxifen (4OH-tamoxifen) (Sigma-Aldrich, MO) was injected directly into the stomach of P5 pups, which is easily identified by its milky-white color. 4OH-tamoxifen was dissolved in 100% ethanol at 10 mg/ml and 5 µl/day were administered for 2 days. $Fig4^{-/flox}$,*Hb9Cre* ($Fig4^{-/flox}$,*Mnx1Cre*) mice have been described previously (*Vaccari et al., 2015*). The spontaneous point mutant $VAC14^{L156R}$ is deficient in PIKfyve binding (*Jin et al., 2008*) and was maintained on a C3HeB/FeJ strain background from which the retinal degeneration locus *rd* was removed by repeated backcrossing. $Pikfyve^{flox/flox}$ mice were generated on the C57BL/6J strain background (*Min et al., 2014*) and were crossed with *Olig2Cre/+* mice. $Mag^{-/-}$ mice on a C57BL/6J background have been described elsewhere (*Pan et al., 2005*). LacZ/ EGFP reporter mice (Jackson laboratory stock #003920) were crossed with *Olig2Cre/+* mice.

### Transmission electron microscopy (TEM)

Postnatal day (P)21 and P60-P75 mice were deeply anesthetized with ketamine (200 mg/kg)/xylazine (20 mg/kg body weight) and perfused transcardially with ice-cold phosphate buffer saline (PBS) for 2 min, followed by 4% paraformaldehyde (PFA) and 2.5% glutaraldehyde in Sorensen's buffer and embedded in epoxy resin as described (*Winters et al., 2011*). Semi-thin sections were stained with toluidine blue for light microscopy. TEM micrographs were taken at 10,500–13,500x magnification with a Philips CM-100 or a JEOL 100CX microscope and analyzed using FIJI software. $Fig4^{-/flox}$, *Hb9Cre* (*Vaccari et al., 2015*), $Fig4^{-/flox}$,*Olig2Cre* and $Fig4^{-/flox}$,*SynCre* conditional mutants were analyzed and compared to littermate controls. Throughout the study, control mice are defined as mice that have at least one intact copy of the *Fig4* allele and include the following genotypes (i) $Fig4^{+/-}$, (ii) $Fig4^{-/flox}$, (iii) $Fig4^{+/flox}$,*Olig2Cre* and (iv) $Fig4^{+/flox}$,*SynCre*.

### Immunohistochemistry

Mice between P10 and adulthood were perfused transcardially with ice-cold 4% PFA in PBS. Brains were post-fixed in perfusion solution for 2 hr at 4°C for in situ hybridization. For immunofluorescence labeling, brains were postfixed overnight and cryoprotected in 30% sucrose in PBS. For FluoroMyelin staining, brains were cryosectioned at 25–40 µm. Free-floating sections were rinsed 3x 5 min in PBS and then stained with FluoroMyelin Green (Millipore, MA, 1:200) in PBS for 20 min. Sections were washed with PBS, mounted onto microscope slides, coverslipped with Prolong Gold antifade supplemented with DAPI (Life Technologies, CA) and imaged with an Olympus IX71 microscope attached to a DP72 camera. For immunofluorescence labeling of optic nerves, nerves were rapidly dissected, kept in perfusion solution for 30 min and cryoprotected in 30% sucrose in PBS. Cross sections (12–20 µm) were mounted onto microscope slides, rinsed 3x for 5 min in PBS and incubated for 1 hr in blocking solution: 1% horse serum and 0.1% Triton-X100 in PBS (anti-Olig2) or 4% normal goat serum and 0.3% Triton-X100 in PBS (anti-NG2). Primary antibody incubation was done overnight at 4°C in blocking solution with rabbit anti-Olig2 (1:1000 Millipore) or rabbit anti-NG2 (1:800, Abcam, UK). The next day, sections were rinsed 3x 5 min with PBS, incubated with appropriate secondary antibodies for 1 hr at room temperature (1:1000, Alexa-conjugated, Life technologies), rinsed in PBS and mounted in Prolong Gold supplemented with DAPI.

## RNA in situ hybridization

cDNA fragments of *Mbp and Plp1* (*Ye et al., 2009*) were used to produce digoxigenin-labeled cRNA probe by run-off in vitro transcription. Brains were cryosectioned at 25 μm and mounted directly onto Superfrost$^+$ microscope slides (Fisher Scientific, MA). Optic nerve sections were prepared as described above and postfixed in 4% PFA/PBS overnight at 4°C. The following day, sections were rinsed with 1x PBS and dehydrated with series of ethanol dilutions (50%, 70%, 95%, and 100%). Sections were then treated with 50μg/ml proteinase K in PBS/5mM EDTA for 15 min (optic nerves) and 30 min for brain sections. All subsequent steps were performed as described previously (*Winters et al., 2011*).

## Isolation of brain membranes

P21 mouse brains were homogenized in a Wheaton Dounce tissue homogenizer cooled on ice. Brain membranes were isolated by centrifugation in a discontinuous sucrose gradient as described previously (*Winters et al., 2011*).

## Isolation of brain tissue

P21 control littermate and *Fig4$^{-/flox}$,PdfrαCreER* brains were extracted and rapidly dissected on ice. Tissue was separated into two groups: 1) cerebellum + brainstem and 2) neocortex + hippocampus + thalamus ('forebrain'). Tissue was lysed in a radio-immunoprecipitation assay buffer (RIPA) using a tissue homogenizer and triturated with a 16G needle. Lysates were spun at 14,000 rpm for 15 min at 4°C and supernatants were analyzed by Western blotting as described below.

## Western blot analysis

Equal amounts of protein (7.5–15 μg) from brain membranes were separated by SDS-PAGE and transferred onto PVDF membranes (Millipore). Membranes were blocked in 3% dry milk powder dissolved in Tris-HCl pH 7.4 buffered saline containing 0.3% Triton X-100 for at least 1 hr and incubated with primary antibody overnight at 4°C. Primary antibodies included mouse anti-βIII tubulin (1:20,000; Promega, WI), rabbit anti-MAG (1:1000; *Winters et al., 2011*), rat anti-MBP (1:1000; Millipore), mouse anti-CNPase (1:1000, Abcam), anti-PLP (1:1000, Abcam), and mouse anti-Fig4 (1:200, NeuroMab, CA). Primary antibodies were detected using either horseradish peroxidase (HRP)-conjugated secondary antibodies (1:2000–15000; Millipore Bioscience Research Reagents) or Alexa-conjugated secondary antibodies (1:20,000, Molecular Probes). The Licor C-DiGit and Odyssey imaging systems and software were used for visualization and quantification of protein bands (Licor, NE).

## Electrophysiology

Recordings were carried out as described elsewhere (*Carbajal et al., 2015*). Briefly, juvenile (P21-P23) and adult (3–4 months) mice were sacrificed by $CO_2$ inhalation. Optic nerves were rapidly dissected, incubated at room temperature in oxygenated artificial cerebrospinal fluid (ACSF) for 45 min and then transferred to a temperature-controlled recording chamber (held at 37 ± 0.4°C) with oxygenated ACSF. Each end of the nerve was drawn into the tip of a suction pipette electrode. The stimulating electrode was connected to a constant-current stimulus isolation unit (WPI, FL) driven by Axon pClamp 10.3 software and a 50 μs pulse was applied to the retinal end of the nerve. The recording electrode was applied to the chiasmatic end of the nerve and connected to the input of a differential AC amplifier (custom-made). A second pipette, placed near the recording pipette but not in contact with the nerve, served to subtract most of the stimulus artifact from the recordings. Signals were digitized at 100 kHz through a data acquisition system (Axon Digidata 1440A, Axon pClamp 10.3, Molecular Devices, CA).

## Primary OL cultures and immunocytochemistry

OPCs were isolated from P6-14 mouse pups with the following genotypes (i) *Fig4$^{+/+}$*, (ii) *Fig4$^{+/-}$*, (iii) *Fig4$^{-/-}$*, (iv) *Fig4$^{-/flox}$,Olig2Cre* or (v) *Pikfyve$^{flox/flox}$,Olig2Cre*. For immunopanning, anti-PDGFRα (BD Biosciences, CA) or O4 antibody (hybridoma cells kindly provided by Dr Jonah Chan) coated plates were used, as described (*Emery and Dugas, 2013* ). For the first two days in vitro, OPCs were cultured on poly-D-lysine (Sigma-Aldrich) coated glass coverslips in DMEM-SATO medium

supplemented with forskolin (Sigma, 10 ng/ml), PDGF (20 ng/ml, Peprotech, NJ), CNTF (10 ng/ml, Peprotech), and NT3 (1 ng/ml, Peprotech). For differentiation studies, OPCs were switched to medium supplemented with T3 (40 ng/ml, Sigma-Aldrich) without growth factors. Cells were allowed to differentiate for 4–6 days prior to fixation in 4% PFA/PBS at RT for 15 min. For immunofluorescence labeling, cells were rinsed 3x 5 min each in PBS, permeabilized with 0.1% Triton-X100 in PBS for 30 min and blocked for 60 min in 3% BSA in PBS. The following primary antibodies were used: rabbit anti-NG2 (1:500, Millipore), rat anti-PDGFRα (1:1000, BD Biosciences, CA), rabbit anti-PDGFRα (1:500, Cell Signaling, MA), rat anti-MBP (1:300, Millipore), rabbit anti-CNPase (1:1000, Assay Biotech, CA), rabbit anti-Ki67 (1:1000, Abcam), mouse anti-MAG (1:300, Millipore), rat anti-Lamp1 (1:1000, Abcam), mouse anti-GFAP (1:2000, Sigma-Aldrich). Cells were incubated with primary antibodies overnight at 4°C. The following day, cells were rinsed 3x 5 min each with PBS, and incubated with secondary antibodies for 1 hr in blocking solution. Following several rinses in PBS, cells were incubated with the nuclear markers Hoechst 33,342 or ToPro3 dye (Life Technologies) and imaged with an Olympus IX71 inverted microscope (Olympus, JP) with a DP72 camera or a Leica SP5 confocal microscope (Leica, DE). Representative confocal images were taken at 63x magnification as z-stacks with 1 μm intervals. Maximum intensity z projections were generated using Fiji. For cell viability experiments, the Live/Dead kit (Life Technologies) was used following the manufacturer's instructions. For actin staining, Actin Red 555 (Life Technologies) was used following the manufacturer's instructions.

For live cell imaging, OPCs were switched to T3 supplemented differentiation medium and kept at 37°C in a 5% $CO_2$ incubator equipped with an IncuCyte Zoom imaging system (Essen Bioscience, MI). Images were taken with a 20x objective every 2 hr for 3 days. Data were analyzed using the IncuCyte Zoom software and Fiji.

## Live cell imaging

O4[+] primary OLs were isolated by immunopanning as described above and cultured in 35 mm glass bottom dishes (Mattek, MA). After 2–3 days under differentiation conditions, anti-MAG-Alexa488 conjugated antibody (1:500, Millipore, MAB1567A4) was added to the culture medium for 12–14 hr. The following day, LysoTracker Deep Red (1:2000, Life Technologies) was added to the culture medium for 30–45 min. Fifteen minutes before imaging, the culture medium was replaced by 1x HBSS (Life Technologies) containing Prolong Live Antifade reagent (Life Technologies, 1:100) and Hoechst dye 33,342 (1:50,000) or NucRed Live 647 (Life Technologies). Cells were imaged at 37°C and ambient $CO_2$ for 15–20 min/dish using a Leica SP5 confocal microscope. Confocal Z-stacks, xyt, and xyzt videos were acquired. As a specificity control for the anti-MAG-Alexa488 antibody, OLs were prepared from $Mag^{-/-}$ and age-matched $Mag^{+/+}$ pups and imaged under identical conditions. Mouse monoclonal anti-MOG antibody (Millipore) was conjugated with Alex555 using the Antibody Labeling Kit (Life Technologies). Some OL cultures were incubated with anti-MOG-Alexa555 (1:250) and anti-MAG-Alexa488 as described above. To some cultures 1 μM apilimod (Axon 1369; Axon Medchem BV) in DMSO was added 90–120 min prior to imaging. Images and videos were processed using Leica AS LF and Fiji. Tracking and movement analysis of anti-MAG-Alexa488[+] particles in live cells was performed using Imaris (Bitplane, UK).

## Ex vivo MAG labeling

To monitor MAG trafficking in acute brain tissue, sagittal slices were prepared from P13-P14 pups with the following genotypes, (i) *Pikfyve* control mice, (ii) littermates *Pikfyve* $^{flox/flox}$,*Olig2Cre* mice, (iii) $Mag^{-/-}$, mice and (iv) P18 *LacZ/EGFP, Olig2Cre* reporter mice (*Toth et al., 2013*). Briefly, mice were decapitated, brains rapidly dissected and submerged in ice-cold ACSF (*Toth et al., 2013*). From forebrain tissue, hippocampi were removed and discarded. Cortex and striatum were sectioned at 300 μm using a tissue slicer (WPI, FL). Brain slices were kept in oxygenated (95% $O_2$, 5% $CO_2$) ACSF at RT for 40–60 min prior to incubation with anti-MAG-Alexa-555 (1:500) in oxygenated ACSF at 32°C for 2 hr. Brain slices were then fixed in 4% PFA for 25 min, rinsed 3 times for 10 min each in PBS and incubated overnight with a goat anti-mouse Alexa-488 secondary antibody (1:1000) in 3% BSA at 4°C. The following day, slices were rinsed 3 times for 10 min each in PBS, incubated with LiveRed 647 for 25 min at RT, rinsed 3 times for 10 min each in PBS, and mounted in Prolong

antifade with DAPI. Individual MAG+ cells in deep cortical layers and striatum were imaged using a Leica SP5 confocal microscope.

## Primary OL transfection

For transfection of primary OPC/OLs, Lipofectamine2000 (Life Technologies) was used, following a protocol previously established for transfection of primary neurons (*Duan et al., 2014*). Briefly, 250 ng of *LAMP1-mCherry* or *Rab7-YFP* plasmid DNA were combined with 1 µl of Lipofectamine2000 (Invitrogen, CA) in optiMEM and mixed thoroughly. Transfection solution was added to OL culture medium and cells were incubated for 2.5 hr. Afterwards, the medium was completely replaced with fresh T3 supplemented medium. To visualize MAG trafficking, anti-MAG-Alexa-488 antibody was added to the culture medium as described above. The following day, live imaging of LAMP1-mCherry+/anti-MAG-Alexa-488+ OLs was carried out as described above.

## Western blot analysis of OPC cultures

OPCs were allowed to expand in PDGF supplemented culture medium for 7–8 days, passaged and plated in 6-well culture dishes at a density of 200,000–300,000 cells/well and kept for 3 days in T3 supplemented medium. Cells were then processed for Western blotting as previously described (*Raiker et al., 2010*). Capillary immunoassays were performed using the automated Wes system (ProteinSimple, San Jose CA). All procedures were performed according to manufacturer's protocol. In brief, 0.8 µg of lysate (4 µl) were mixed with 2 µl of 5x fluorescent master mix and boiled for 5 min. These samples were dispensed into microplates along with blocking solution, primary and secondary antibodies and chemiluminescent substrate. After centrifugation, microplate was loaded into the Wes instrument for subsequent protein separation on capillaries and immuno-detection using the standard electrophores, immunolabeling, detection scheme of Wes. Data were analyzed by using Compass software (ProteinSimple) and peak areas were used for quantification. Erk1 peak area was used for normalization between samples. Three independent preparations were processed.

## Statistical analysis

To assess myelination in the optic nerve, ten non-overlapping TEM images were randomly selected and the fraction of myelinated axons quantified as described (*Winters et al., 2011*). At least 600 axons were quantified per nerve. G-ratio analysis was performed as described previously (*Winters et al., 2011*). At least 100 axons per optic nerve were analyzed. For Western blot analysis, Western band intensity was measured using LI-COR Studio Image Software. All band intensities were normalized either to βIII-tubulin (brain lysates and membranes) or actin (OPC cultures). Normalized Western blot band intensity for control samples was set as 1 for each experiment. For optic nerve electrophysiology, data analysis was performed offline using Clampfit software. In order to analyze individual peaks, each trace was fitted as a sum of three or four Gaussians using Origin Pro software (*Chen et al., 2004*). A peak with the largest amplitude in each trace was used for conduction velocity analysis.

For quantification of *Plp1*, Olig2, and NG2 labeled cells, the number of respective positive cells was quantified per optic nerve cross section and normalized to the section area (arbitrary units in FIJI). At least four sections per nerve were analyzed.

For quantification of OL markers in vitro, ten non-overlapping images were taken at random positions for each coverslip/well and cells positive for a marker of interest counted and normalized to the number of Hoechst 33,342 dye positive cells in the same image. A minimum of 900 cells was quantified for each individual experiment with *Fig4* cultures and a minimum of 120 cells was quantified for each individual experiment with *Pikfyve* cultures. GFAP+ astrocytes were excluded from quantification. The analysis of actin/MBP postmitotic OL morphology was performed as characterized previously (*Zuchero et al., 2015*).

For cell viability experiments, the Live/Dead kit was used the number of live (green) and dead (red) cells was quantified and the live/total cell ratio was calculated. For all experiments, Hoechst 33,342 normalized cell density in control groups was set as 1. At least three independent experiments with duplicate coverslips were used for the analysis. For live imaging of MAG+ vesicles in primary OLs, Imaris software (Bitplane) was used to calculate individual particle speed and size. Four

independent experiments were analyzed for $Fig4^{+/+}$ and $Fig4^{-/-}$ cultures. $MAG^+$ particles of at least 0.01 µm³ in volume were included in data analysis.

One-way ANOVA followed by Tukey posthoc was used for TEM optic nerve analysis. One-way ANOVA followed by Dunnett's posthoc was used for Western blot analysis and electrophysiology with more than two groups. The unpaired Student t-test was used for analysis in all experiments with two groups.

## Acknowledgements

We would like to thank members of Giger lab for their feedback on manuscript preparation, Ben Barres and Ronald Schnaar for providing *Olig2Cre* and *Mag*⁻/⁻ mice, respectively, Bradley Zuchero for his advice on OPC immunopanning, Jonah Chan for providing O4 hybridoma cells, Bill Tsai and Takamasa Inoue for the Rab7-YFP plasmid, Richard Lu for the Plp1 plasmid, and Lois Weisman for apilimod We thank Margaret Youngman for assistance with optic nerve electrophysiology and Brian Pierchala and Jennifer Shadrach for help with confocal microscopy.

## Additional information

### Funding

| Funder | Grant reference number | Author |
|---|---|---|
| National Institute of Child Health and Human Development | T32HD007505 | Yevgeniya A Mironova |
| National Institute of General Medical Sciences | T32GM007315 | Yevgeniya A Mironova |
| Dr. Miriam and Sheldon G. Adelson Medical Research Foundation | APNRR | Jeffery L Twiss Leif A Havton Roman J Giger |
| National Heart, Lung, and Blood Institute | HL040387 | Charles S Abrams |
| National Heart, Lung, and Blood Institute | HL120846 | Charles S Abrams |
| National Institute of Neurological Disorders and Stroke | R01NS081281 | Peter Shrager Miriam H Meisler Roman J Giger |
| Schmitt Program on Integrative Brain Research | | Peter Shrager |
| National Institute of General Medical Sciences | R01GM24872 | Miriam H Meisler |

The funders had no role in study design, data collection and interpretation, or the decision to submit the work for publication.

### Author contributions

YAM, Conception and design, Acquisition of data, Analysis and interpretation of data, Drafting or revising the article; GML, SJL, LAH, Acquisition of data, Analysis and interpretation of data; J-PL, Acquisition of data, Analysis and interpretation of data, Drafting or revising the article; JLT, Analysis and interpretation of data, Drafting or revising the article, Contributed unpublished essential data or reagents; IV, SHM, Acquisition of data, Contributed unpublished essential data or reagents; AB, CSA, Drafting or revising the article, Contributed unpublished essential data or reagents; PS, Analysis and interpretation of data, Drafting or revising the article; MHM, RJG, Conception and design, Analysis and interpretation of data, Drafting or revising the article

### Author ORCIDs

Roman J Giger, http://orcid.org/0000-0002-2926-3336

### Ethics

Animal experimentation: This study was performed in strict accordance with the recommendations in the Guide for the Care and Use of Laboratory Animals of the National Institutes of Health. All of the animals were handled according to protocols approved by the University committee on use and care for animals (UCUCA protocols: #00005863 and #00005902) of the University of Michigan.

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
