## [Decision Letter]

Thank you for submitting your work entitled "PI(3,5)P_2_ Biosynthesis Regulates Oligodendrocyte Differentiation by Intrinsic and Extrinsic Mechanisms" for consideration by *eLife*. Your article has been reviewed by two peer reviewers and the evaluation has been overseen by a Reviewing Editor and Gary Westbrook as the Senior Editor.

The reviewers have discussed the reviews with one another and the Reviewing Editor has drafted this decision to help you prepare a revised submission.

Summary:

Here, the authors investigate trafficking mechanisms involving PI(3,5)P_2_ in OLs and neurons and how this trafficking pathway regulates myelination. The starting point is that *Fig4*-deficient mice have profound hypomyelination and spongiform neuron degeneration. The authors further investigate the mechanisms of FIG4 function by generating mice lacking *Fig4* specifically in neuronal or oOLsodendrocyte lineages. They find the following major results: 1) loss of *Fig4* in OLs myelins OL maturation, widespread hypomyelination, and the aberrant trafficking of myelin proteins. In vitro large intracellular vesicles accumulate. 2) Loss of *Fig4* in neurons also causes hypomyelination, reduced OL maturation, but in addition it causes more severe neurodegeneration. As expected, phenotypic analysis of these mice shows hypomyelination by EM, reduced conduction velocities, etc. Overall, the data are interesting and sound. While the first half of the paper deals with the phenotypic analysis of the mice (in keeping with the previous publications), the second half of the paper really breaks new ground as the authors show that mice lacking PIKfyve enzymes or are mutant for *Vac14* also have similar hypomyelination and OL maturation defects. These experiments are very nice and provide additional confirmation of the importance of the Fig4/Vac14/PIKfyve enzyme complex for normal myelination. The authors then show the hypomyelination defects (not associated with OL maturation) may result from impaired trafficking of myelin proteins like MAG and PLP.

Essential revisions:

1) No g ratio analysis is shown to make it clear whether *Fig4* null oligodendrocytes form normal myelin sheath.

2) The paper demonstrates that absence of other components components of the *Fig4* complex, PIKFYVE and VAC14 may also lead to hypomyelination, although the data here are in a way premature and should include convincing morphological analysis.

3) Altogether, the data supports a role for the PI(3,5)P_2_ biosynthesis complex in myelination by regulating membrane trafficking in oligodendrocytes, but additional experiments would make the story more convincing.

4) Although the discovery of the Fig4/Vac14/PIKfyve enzyme complex for normal myelination is significant, these results still don't tell us anything about the nature of what is being trafficked in neurons or immature OLs that leads to the hypomyelination phenotypes. Of course, this is very hard to do and will no doubt be the focus of the next paper.

---

## [Author Response]

Essential revisions:

1) No g ratio analysis is shown to make it clear whether Fig4 null oligodendrocytes form normal myelin sheath.

We agree with the reviewers’ comment and have included g-ratio analysis for the *Fig4* conditional knock-out (cKO) mice lacking *Fig4* in neurons (*SynCre*) or in the OL lineage (*Olig2Cre*) mice. In both cKO mice, g ratios are increased, indicating that myelin sheath thickness is reduced. These data are now shown in Figure 2.

2) The paper demonstrates that absence of other components of the Fig4 complex, PIKFYVE and VAC14 may also lead to hypomyelination, although the data here are in a way premature and should include convincing morphological analysis.

Morphological studies of epoxy resin embedded and toluidine blue stained optic nerve sections from P14 *Pikfyve^flox/flox^, Olig2Cre* mice (Figure 6—figure supplement 1) and *VAC14^L156R/L156R^* (Figure 7—figure supplement 1) are now included in the manuscript. Loss of *Pikfyve* in the OL lineage leads to a complete loss of myelin. Consistent with electrophysiological and biochemical studies, toluidine blue labeling of *VAC14^L156R/L156R^* optic nerve sections revealed severe hypomyelination.

3) Altogether, the data supports a role for the PI(3,5)P2 biosynthesis complex in myelination by regulating membrane trafficking in oligodendrocytes, but additional experiments would make the story more convincing.

We agree with the reviewers’ comment. In addition to the membrane trafficking studies in primary OLs, shown in Figure 8 and Figure 9 of the original submission, we now included OL membrane trafficking studies in acute forebrain tissue from P14 mice (Figure 10 and Figure 10—figure supplement 1). We show that in acute wildtype brain slices (kept in oxygenated artificial cerebrospinal fluid) plasma membrane derived MAG is rapidly endocytosed and localized to small vesicular structures that accumulate in the cell soma and OL processes that form internodes. To visualize MAG trafficking, brain slices were incubated with mouse anti-MAG-Alexa555 antibody for two hours. To distinguish between surface-localized MAG and endocytosed MAG, brain slices were fixed and incubated with an anti-mouse Alexa488 secondary antibody under non-permabilizing conditions. As shown in Figure 10”, the majority of MAG in wildtype slices is endocytosed and only labeled by anti-MAG-Alexa555. To demonstrate antibody specificity, parallel studies with *Mag^-/-^*brain slices were carried out (Figure 10—figure supplement 1) and revealed no staining above background. To demonstrate anti-MAG-Alexa555 antibody is taken up by OLs, cells in the OL lineage were genetically labeled (*LacZ/EGFP, Olig2Cre).* Three-dimensional rendering of EGFP^+^ cells (OLs) revealed accumulation of MAG in small intracellular vesicles (see Insert in Figure 10—figure supplement 1). Thus far these experiments show that we can monitor MAG endocytosis and vesicular localization in acute brain slices of wildtype mice. To demonstrate a role for the lipid PI(3,5)P_2_ in this process, experiments were repeated with acute brain slices from *Pikfyve^flox/flox^, Olig2Cre* mice. As shown in Figure 10”, MAG trafficking is impaired. Endocytosis of MAG from the plasma membrane does occur in some cells (see insert in Figure 10”), vesicle accumulation is perinuclear and no labeling of endocytosed MAG in cellular processes is observed. Together, these studies show that PI(3,5)P_2_ plays a critical role in mobilization of plasma membrane derived MAG in brain slices.

4) Although the discovery of the Fig4/Vac14/PIKfyve enzyme complex for normal myelination is significant, these results still don't tell us anything about the nature of what is being trafficked in neurons or immature OLs that leads to the hypomyelination phenotypes. Of course, this is very hard to do and will no doubt be the focus of the next paper.

These are important questions and the focus of ongoing studies. In the revised manuscript we have added one additional piece of data. We now show that trafficking of MAG through LE/Lys is impaired, as plasma membrane derived MAG accumulates in large perinuclear vesicles in *Fig4^-/-^*OLs. Trafficking of plasma membrane derived MOG occurs through recycling endosomes (RE) and is not affected by *Fig4* deficiency. This shows that PI(3,5)P_2_ is required for membrane trafficking through the LE/Lys compartment but not through RE.